# Retinoic acid alleviates the reduction of Akt and Bad phosphorylation and regulates Bcl-2 family protein interactions in animal models of ischemic stroke

**Ju-Bin Kang, Phil-Ok Koh**[ID]*

Department of Anatomy, College of Veterinary Medicine, Research Institute of Life Science, Gyeongsang National University, Jinju, South Korea

* pokoh@gnu.ac.kr

**Data Availability Statement:** All relevant data are within the paper and its Supporting Information files.

## Abstract

Ischemic stroke causes a lack of oxygen and glucose supply to brain, eventually leads to severe neurological disorders. Retinoic acid is a major metabolic product of vitamin A and has various biological effects. The PI3K-Akt signaling pathway is an important survival pathway in brain. Phosphorylated Akt is important in regulating survival and apoptosis. We examined whether retinoic acid has neuroprotective effects in stroke model by regulating Akt and its downstream protein, Bad. Moreover, we investigated the relationship between retinoic acid and Bcl-2 family protein interactions. Animals were intraperitoneally administered vehicle or retinoic acid (5 mg/kg) for four days before surgery and ischemic stroke was induced by middle cerebral artery occlusion (MCAO) surgery. Neurobehavioral tests were performed 24 h after MCAO and cerebral cortical tissues were collected. Cresyl violet staining and TUNEL histochemistry were performed, Western blot and immunoprecipitation analysis were performed to elucidate the expression of various proteins. Retinoic acid reduced neurological deficits and histopathological changes, decreased the number of TUNEL-positive cells, and alleviated reduction of phospho-PDK1, phospho-Akt, and phospho-Bad expression caused by MCAO damage. Immunoprecipitation analysis showed that MCAO damage reduced the interaction between phospho-Bad and 14-3-3, which was attenuated by retinoic acid. Furthermore, retinoic acid mitigated the increase in Bcl-2/Bad and Bcl-xL/Bad binding levels and the reduction in Bcl-2/Bax and Bcl-xL/Bax binding levels caused by MCAO damage. Retinoic acid alleviated MCAO-induced increase of caspase-3 and cleaved caspase-3 expression. We demonstrate that retinoic acid prevented apoptosis against cerebral ischemia through phosphorylation of Akt and Bad, maintenance of phospho-Bad and 14-3-3 binding, and regulation of Bcl-2 family protein interactions.

**Funding:** This research was supported by the National Research Foundation of Korea (NRF) grant funded by the Korea government [MEST][NRF-2021R1F1A105878711] and (MSIT) [RS-2023-00248145]. The funders had no role in study design, data collection and analysis, decision to publish, or preparation of the manuscript

**Competing interests:** The authors have declared that no competing interests exist.

## Introduction

Stroke is a serious disease with high mortality and morbidity and is a major cause of human death [1]. The ischemic stroke is the main type of stroke, which is caused by blocked blood vessels and is also called cerebral infarction. Ischemic stroke blocks blood flow to the brain, resulting in a lack of oxygen and glucose supply [2]. It causes harmful changes in biochemical processes such as oxidative stress generation, ATP production reduction, neuroinflammatory response, and excitatory toxicity, which eventually cause irreversible brain damage [3, 4].

Retinoic acid is a representative bioactive derivative of vitamin A and performs a variety of biological functions including antioxidant, anti-inflammatory, and anti-apoptotic functions [5–7]. Retinoic acid is associated with brain development such as neuronal differentiation and axonal growth [8, 9]. It performs neuroprotective effects by regulating neuroinflammatory and neurodegenerative mechanisms [10]. Retinoic acid reduces cerebral infarction volume in focal cerebral ischemia and protects neurons [11]. In addition, retinoic acid has the advantage of easily passing the blood-brain barrier (BBB) and alleviates BBB disruption following ischemic stroke. Furthermore, low circulation level of retinoic acid is associated with increased risk of death. Thus, retinoic acid is considered to play an important role in neurological functions in the adult brain as well as in development of the central nervous system.

Akt, also known as protein kinase B, is a serine/threonine-specific protein kinase that plays an important role in regulating survival and cell death [12]. Akt is involved in the regulation of various processes, including glucose metabolism, cell proliferation, and cell death [13, 14]. Various growth and survival factors activate the phosphatidyl-inositol-3-kinase (PI3K)/Akt signaling pathway. Akt is phosphorylated directly at the threonine 308 site by phosphoinositide-dependent protein kinase 1 (PDK1) [15]. Akt inhibits apoptosis and promotes cell survival through phosphorylation of proapoptotic proteins including Bad, forkhead transcription factors (FKHR), and glycogen synthase kinase-3β (GSK-3β) [16, 17]. Among these proteins, Bad is accepted as a representative pro-apoptotic Bcl-2 family proteins. Bad promotes apoptotic cell death by binding with anti-apoptotic proteins such as Bcl-2 and Bcl-xL. However, phosphorylation of Bad dissociates Bad from the Bcl-2/Bad or Bad/Bcl-xL complex, prevents Bad interaction with Bcl-2 or Bcl-xL, and activates anti-apoptotic functions of Bcl-2 and Bcl-xL to inhibit apoptosis [18]. Phosphorylated Bad prevents pro-apoptotic processing of Bad by binding to the chaperone protein 14-3-3 [19–21]. However, in the absence of growth or survival factors, Bad interacts with Bcl-2 or Bcl-xL and dissociates Bax from the Bcl-2/Bax and Bcl-xL/Bax complexes [22]. Activated Bax releases cytochrome *c* from mitochondria to the cytosol and continuously activates the caspase cascade, which results in apoptotic cell death [23]. The PI3K-Akt-Bad signaling pathway is important for cell survival, and phosphorylation of Bad is critical for cell survival.

Ischemic stroke induces cell excitotoxicity, mitochondrial dysfunction, blood-brain barrier damage, neuroinflammation, and apoptotic processes. It is also known that the mechanism of causing ischemic stroke is very complicated. Moreover, the signaling pathway of ischemic stroke is very diverse and complex. The signal pathways involved in stroke are PI3K/Akt signaling pathway, phosphatase and tensin homolog signaling pathway, death-associated protein kinase 1 signaling pathway, neuronal nitric oxide synthase signaling pathways, hypoxia-inducible factor signaling pathway, nuclear factor E2-related factor 2 signaling pathway, casein kinase 2 signaling pathway, mTOR-related signaling pathways, and p53-mediated apoptotic pathway [24–32]. As mentioned above, stroke is associated with various signaling pathways, but it is difficult to discuss all pathways in this study. Among these various signaling pathways, we focused on the PI3K/Akt signal pathway, which is a representative survival pathway. In previous studies, retinoic acid activated the Akt signaling pathway and promoted cell survival in

various tissues [5, 33, 34]. It has also been shown to enhance neural differentiation by upregulating DAX1 levels through the PI3K-Akt pathway [35]. Ischemic stroke is a complex neurological disorder in which signaling pathways are disrupted [36]. PI3K-Akt pathway is a representative signaling pathway involved in ischemic stroke [36]. Moreover, recent study has focused on the potential therapeutic significance through PI3K-Akt pathway activation in ischemic stroke [37]. However, data on the neuroprotective effects of retinoic acid on the PI3K-Akt signaling pathway in cerebral ischemia are limited. It has not been previously reported whether retinoic acid regulates the expression of phospho-Bad and the interaction between phospho-Bad and 14-3-3 in a stroke animal model. Therefore, this study was designed to investigate the neuroprotective effects of retinoic acid on cerebral ischemia and the regulation of phospho-Akt and phospho-Bad by retinoic acid. In addition, we examined whether retinoic acid inhibits apoptosis and protects brain tissues from cerebral ischemia by controlling the binding of phospho-Bad with 14-3-3 and binding of Bcl-2 and Bcl-xL to Bad or Bax.

## Materials and methods

### Experimental animal preparation

Male Sprague Dawley ($n$ = 40, 210–220 g) rats were obtained from Samtako Co. (Animal Breeding Center, Osan, Korea). Male animals were used to eliminate potential confounding variables associated with sex hormones. All experimental procedures were conducted by following approved guidelines of the Institutional Animal Care and Use Committee of Gyeongsang National University (Approval number: GNU-190218-R0008). Animals were housed with controlled temperature condition with 25°C and lighting condition with 12 h light and 12 h dark cycle. They were randomly divided into four groups as follows: vehicle + sham, retinoic acid + sham, vehicle + middle cerebral artery occlusion (MCAO), and retinoic acid + MCAO. Retinoic acid (5 mg/kg, Sigma Aldrich, St. Louis, MO, USA) was dissolved in solvent agent (polyethylene glycol, 0.9% NaCl, and ethanol; 70%/20%/10% by volume) and injected via intraperitoneal cavity four days before surgery [38]. Animals were treated with 5 mg/kg of retinoic acid, and the dose and duration of treatment of retinoic acid were determined by the previously described method [38]. We previously confirmed the neuroprotective effect of retinoic acid at this dose and duration of retinoic acid administration [39]. Vehicle group animals were injected only solvent solutions.

### Middle cerebral artery occlusion surgery

MCAO surgery was performed to induce cerebral ischemia in the following method [40]. Animals were anesthetized by intraperitoneal injection with Zoletil (50 mg/kg, Virbac, Carros, France) 30 min after retinoic acid injection and kept in a heating pad in a supine position to maintain body temperature. Right common carotid artery (CCA) was exposed through the midline cervical incision and separated from adjacent tissues. External carotid artery (ECA) and internal carotid artery (ICA) were exposed. CCA was temporarily ligated using microvascular clamps, the laryngeal artery and cranial thyroid artery were resected, and the ECA was amputated. A 4/0 monofilament with heat-rounded tip was inserted into the cut ECA, continuously moved to the ICA until resistance was felt to block the origin of the middle cerebral artery, and ligated with ECA. Microvascular clips were removed and incised skin was sutured. Middle cerebral artery was occluded for 24 h [38, 39, 41]. Sham animals were operated the same procedure except for insertion of nylon filament. Neurobehavioral tests including neurological deficits scoring test, corner test, and grip strength test are commonly used in rodent models of stroke [42–44]. Animals were assessed on neurobehavioral tests 24 h after surgery. After anesthesia with Zoletil (Virbac), animals were quickly decapitated and sacrificed for

experimentation. We tried to minimize pain to the animals and brain tissues were collected for further study.

## Neurological deficits scoring test

Neurobehavioral disability was examined with neurological deficits scoring test [42]. Neurological deficits were scored according to the five-point scale: no recognizable neurological deficits (0); lack of spontaneous motor activity or flexion of the contralateral forelimb (1); instinctive circling to the contralateral side (2); falling to the contralateral side or lack of spontaneous motor activity (3); dysfunctional spontaneous activity (4). Three researchers simultaneously observed behavioral changes and scored each to reduce deviations. The obtained scores were averaged and presented as the final result.

## Corner test

Corner test was performed to test the sensorimotor abnormalities according to following method [43]. Animals were placed in the corner which was made up with two board pieces (30 × 20 × 1 cm). The edges of the board were connected to the open end and the angle was 30˚. When animals reached the corner, both sides of vibrissae were touched and they turned back to the open end. Animals were trained for seven days before MCAO surgery and the ratio of turning left or right side was similar. Ten times were tried and results were expressed as the number of turns. We provided a corner test interval of 1 min for the accuracy of the experiment.

## Grip strength test

Grip strength of the left and right forelimbs were examined using a grip strength meter (Jeung Do Bio & Plant Co., Ltd., Seoul, Korea) and tested according to previously described method [44]. When animal grasped the bar, the force gauge was reset to 0 kg and animal's tail was slowly pulled back. Maximum force tension was recorded. The test was performed with only one forelimb at a time. For example, when examining right forelimb, the other left forelimb was wrapped with masking tape. Each animal was tested five times and average was used as grip strength. The interval of the grip strength test was 5 min.

## Cresyl violet staining

Brain tissues from bregma levels +2.0 mm to -2.0 mm were fixed with 4% paraformaldehyde solution and washed with tap water overnight to remove paraformaldehyde. They were dehydrated with the ethyl alcohol series from 70% to 100% and washed with xylene. They were infiltrated with paraplast® (Leica, Wetzlar, Germany) under vacuum and embedded in paraffin embedding center (Leica). Paraffin tissues were cut to 4 μm thicknesses and paraffin sections were mounted on slide glasses. They were dried on slide warmer (Thermo Fisher Scientific, Waltham, MA, USA) and deparaffinized with xylene. The tissues were hydrated by gradually progressing from 100% ethyl alcohol to 70% ethyl alcohol for 1 min each, and washed with water. Sections were stained with 1% cresyl violet solution (Sigma Aldrich) for 10 min and washed with distilled water. Stained sections were dehydrated by gradually progressing from 70% ethyl alcohol to 100% ethyl alcohol for 1 min each. They were cleared with xylene and mounted with a permount solution (Thermo Fisher Scientific). For histopathological observation, the level of brain slices was selected as bregma 1.20 mm. The level of bregma was determined based on previously reported [45]. They were visualized and photographed by Olympus light microscope (Olympus, Tokyo, Japan). Brain section images were analyzed by

Image J software (National Institutes of Health, Bethesda, MD, USA). Intact area was stained with deep violet color, whereas infarct area was un-stained or stained with light violet color. Infarct volume was calculated according to a percentage value (%) by following formula: infarction area/whole section area × 100. Region of cerebral cortex was randomly selected. The number of damaged neurons was counted in each area using light microscope (Olympus) and analyzed by Image J software (National Institutes of Health) by observer-blinded manner. The value of damaged cells was determined as a percentage of the number of damaged cells to the number of total cells.

## Terminal deoxynucleotidyl transferase dUTP nick-end labeling (TUNEL) assay

TUNEL assay was performed to detect apoptotic cells. ApopTag® peroxidase in situ apoptosis detection kit (Merck, Darmstadt, Germany) was used. Deparaffinized and hydrated sections were rinsed with phosphate buffered saline (PBS), pretreated with proteinase K (20 μg/mL, Thermo Fisher Scientific) for 5 min, quenched in 3% hydrogen peroxide in PBS for 5 min, and washed with PBS. They were incubated with equilibration buffer for 1 h at 4°C and reacted with working strength terminal deoxynucleotidyl transferase enzyme for 90 min at 37°C in humidified chamber. Reacted slides were applied with stop/wash buffer for 10 min and washed with PBS. Washed slides were incubated with anti-digoxigenin conjugate for 1 h at room temperature in humidified chamber and washed with PBS. They were reacted with peroxidase substrate 3,3'-diamino benzidine tetrahydrochloride (DAB, Sigma Aldrich), washed with PBS, and counterstained with Harris' hematoxylin solution (Sigma Aldrich). They were dehydrated with ethyl alcohol series from 70% to 100%, cleared with xylene, and coverslipped with a permount solution (Thermo Fisher Scientific). Coverslipped sections were observed and photographed by Olympus light microscope (Olympus). Five areas of the cerebral cortex region were selected and TUNEL-positive cells were counted in each area. The result value of TUNEL assay was shown as a percentage of the number of TUNEL-positive cells to the number of total cells.

## Western blot analysis

Right cerebral cortices were isolated from brain tissues and stored at -70°C until Western blot analysis was performed. Tissues were homogenized with lysis buffer [1% Triton X-100, 1 mM EDTA in PBS (pH 7.4)] with phenylmethanesulfonylfluoride (PMSF, Sigma Aldrich). They were kept on ice for 1 h and centrifuged at 15,000 g for 1 h at 4°C. Supernatants were collected and protein concentrations were measured using bicinchoninic acid protein assay kit (Pierce, Rockford, IL, USA). Protein samples (30 μg) were loaded on 10% sodium dodecyl sulfate polyacrylamide gel, electrophoresed at 10 mA for 30 min, and constantly electrophoresed at 20 mA for 90 min. Electrophoresed gel was transferred to polyvinylidene fluoride membrane (PVDF, Sigma Aldrich) using semi-dry blotting system (ATTO Co., Tokyo, Japan) at 25 V for 25 min. Transferred membrane was blocked with 5% skim milk solution with Tris-buffered saline containing 0.1% Tween-20 (TBST) and washed three times with TBST for 10 min. They were incubated with following primary antibodies; anti-phospho-PDK1 (Ser 241), anti-phospho-Akt (Thr 308), anti-phospho-Bad (Ser 136), anti-Akt (1: 1,000 dilution, Cell Signaling Technology, Danvers, MA, USA), anti-Bad, anti-14-3-3, and anti-β-actin (1:1,000 dilution, Santa Cruz Biotechnology, Dallas, TX, USA) for overnight at 4°C. Membrane was washed three times with TBST for 10 min and incubated with horseradish peroxide-conjugated anti-rabbit IgG or anti-mouse IgG (1: 5,000, Cell Signaling Technology) for 2 h at room temperature. They were washed three times with TBST for 10 min and incubated with enhanced chemiluminescent

reagents (GE Healthcare, Chicago, IL, USA) to detect immunoreactive protein bands. They were exposed to Fuji medical X-ray film (Fuji Film, Tokyo, Japan) to visualize the bands. The intensity value of the protein bands were analyzed by Image J software (National Institutes of Health) and presented as a ratio of specific protein intensity to β-actin intensity.

## Immunohistochemistry

Paraffin sections were deparaffinized with xylene, hydrated with ethyl alcohol series from 70% to 100%, and washed with tap water. Deparaffinized sections were incubated in 0.01 M sodium citrate buffer (pH 6.0) and were heated by microwave oven for antigen retrieval process. Sections were washed with PBS, reacted in 3% hydrogen peroxide for 20 min, and washed with PBS. They were blocked with 5% normal goat serum for 1 h at room temperature to block the non-specific binding, and were reacted with anti-phospho-Akt or anti-phospho-Bad (1:100, Cell Signaling Technology) for overnight at 4˚C. They washed with PBS and incubated with biotin-conjugated secondary antibody for 2 h at room temperature. Sections were washed with PBS, incubated in avidin-biotin-peroxidase complex (Vector Laboratories Inc, Burlingame, CA, USA), and washed with PBS. They were colored with DAB (Sigma Aldrich), washed with tap water, and counterstained with Harris' hematoxylin solution (Sigma Aldrich). They were dehydrated with ethyl alcohol series from 100% to 70%, cleared with xylene, and mounted with permount solution (Thermo Fisher Scientific). We observed stained sections using Olympus light microscope (Olympus). Five fields of the cerebral cortex region were randomly selected and the number of cells was counted in each region. The result value of reaction levels were expressed as a percentage of the number of phospho-Akt-positive cells or phospho-Bad-positive cells to the number of total cells.

## Immunoprecipitation assay

Immunoprecipitation was performed to examine the interaction of phospho-Bad and 14-3-3 interaction and Bcl-2 family proteins interaction. Protein samples (200 μg) were precleared with protein A/G agarose beads (Santa Cruz Biotechnology) to eliminate nonspecific binding proteins. They were mixed with following antibodies: anti-14-3-3, anti-Bad, and anti-Bax antibody (Santa Cruz Biotechnology) and incubated for overnight at 4˚C with mild shaking. Complexes were precipitated with protein A/G agarose beads for 2 h at 4˚C, washed with radioimmunoprecipitation assay buffer (Sigma Aldrich) with PMSF, and centrifuged at 10,000 g for 1 min. Supernatants were removed and remnant were boiled with sample buffer. They were loaded on 10% sodium dodecyl sulfate polyacrylamide gel, electrophoresed, and transferred to PVDF membrane. Membrane were rinsed with TBST and reacted with following antibodies: anti-phospho-Bad (1:1,000, diluted with TBST, Cell Signaling Technology), anti-Bcl-2, and anti-Bcl-xL antibody (1:1,000, diluted with TBST, Santa Cruz Biotechnology). They were washed with TBST and continuously performed as above described method in Western blot analysis.

## Statistical analysis

All results are presented as the mean ± standard error of means (S.E.M.). The results of each group were compared by two-way analysis of variance (ANOVA) followed by post-hoc Scheffe's test. A value of $p < 0.05$ was considered statistically significant. $^{\#}$ $p < 0.05$, $^{*}$ $p < 0.01$.

## Results

### Neuroprotective effects of retinoic acid in MCAO animals

We found that MCAO caused severe neurological behavioral deficits such as instinctive circulation and lack of movement, and retinoic acid treatment alleviated the changes. However, in sham-operated animals, neurobehavioral disorders did not occur regardless of vehicle or retinoic acid treatment. Neurological deficit scores were 3.30 ± 0.15 in vehicle + MCAO animals and 1.80 ± 0.13 in retinoic acid + MCAO animals (Fig 1A). The corner test results showed a preference of response by stimuli. The number of right turns was 9.20 ± 0.20 and 6.50 ± 0.17 in vehicle + MCAO and retinoic acid + MCAO animals, respectively (Fig 1B). The number of left turns was 0.80 ± 0.20 in vehicle + MCAO animals and 3.50 ± 0.17 in retinoic acid + MCAO animals. However, sham-operated animals responded almost identically to the right or left stimuli. Grip strength of the contralateral forelimb was significantly decreased in MCAO animals compared with sham animals. Retinoic acid administration improved MCAO-induced grip loss. In the left forelimb, grip strength was 0.13 ± 0.02 kg in vehicle + MCAO animals and 0.29 ± 0.02 kg in retinoic acid + MCAO animals (Fig 1C). Grip strength of the right forelimb was 0.51 ± 0.02 kg and 0.58 ± 0.02 kg in vehicle + MCAO and retinoic acid + MCAO animals, respectively. In cresyl violet-stained brain tissues, we visually observed the pale areas of the right cerebral cortex of vehicle + MCAO animals, which was the ipsilateral side of MCAO damage (Fig 1D). However, retinoic acid treatment alleviated the area of pale lesions compared with vehicle + MCAO animals. Intact regions were completely stained dark purple, and the ischemic regions were not stained or stained light purple. Extensive ischemic areas were observed in vehicle + MCAO animals, retinoic acid treatment decreased the ischemic region (Fig 1D). Ischemic areas were 24.28 ± 2.19% and 7.24 ± 0.63% in vehicle + MCAO and retinoic acid + MCAO animals, respectively (Fig 1E). The microscopic observation results showed the histopathological changes caused by MCAO damage (Fig 1F). Sham-operated animals had normal neurons that included a pyramidal shape with large and round nuclei. We observed serious histopathological changes including pyknotic nuclei, cytoplasmic vacuolation, and shrunken dendrites in vehicle + MCAO animals. These changes were mitigated in retinoic acid + MCAO animals. We counted the number of damaged cells with abnormal forms and observed the increase in the number of damaged cells in vehicle + MCAO animals. These increases were alleviated in retinoic acid + MCAO animals. The number of damaged cells was 87.74 ± 2.36% in vehicle + MCAO and 40.68 ± 1.34% in retinoic acid + MCAO animals (Fig 1E). TUNEL assay was performed to confirm MCAO-induced apoptosis and TUNEL-positive cells were observed in the cerebral cortex of MCAO animals. The number of TUNEL-positive cells was significantly increased in the vehicle + MCAO animals, retinoic acid treatment attenuated this increase (Fig 1G). The number of TUNEL-positive cells was 81.43 ± 3.15% and 24.60 ± 2.39% in vehicle + MCAO and retinoic acid + MCAO animals, respectively (Fig 1E).

### Modulation of phospho-Akt and phospho-Bad expression by retinoic acid in MCAO animals

The expression of PDK1, phospho-PDK1, Akt, phospho-Akt, Bad, and phospho-Bad was investigated using Western blot analysis. Phospho-PDK1, phospho-Akt, and phospho-Bad expression was decreased in vehicle animals with MCAO damage, retinoic acid treatment alleviated these decreases (Fig 2A). Phospho-PDK1 levels were 0.29 ± 0.02 in the vehicle + MCAO animals and 0.63 ± 0.03 in the retinoic acid + MCAO animals (Fig 2B). Phospho-Akt levels were 0.29 ± 0.02 and 0.68 ± 0.04 in vehicle + MCAO and retinoic acid + MCAO animals,

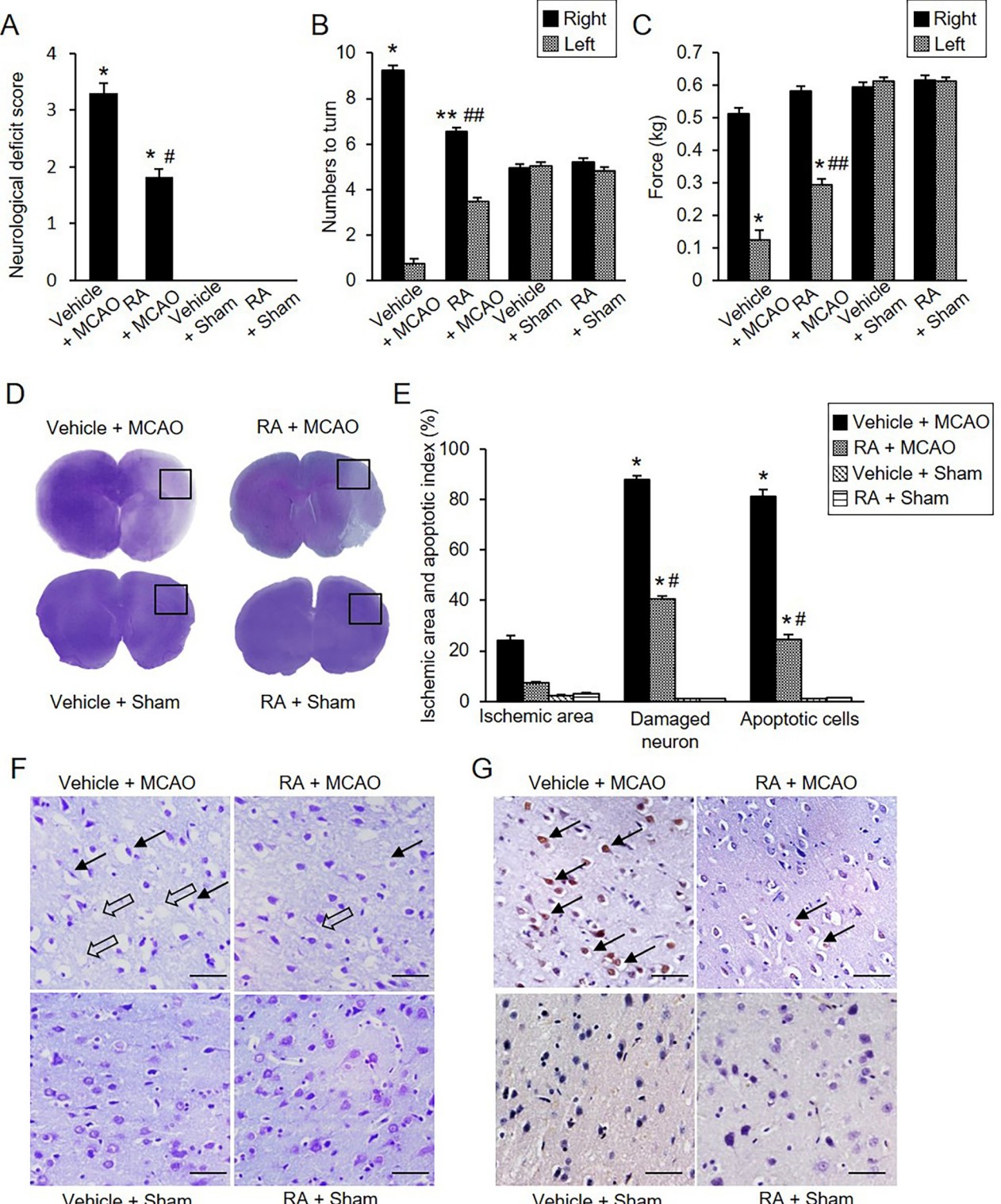

**Fig 1. Retinoic acid improves neurobehavioral disorders caused by MCAO damage.** Neurological deficits scoring test (A), corner test (B), grip strength test (C), gross photographs (D), microscopic photographs of cresyl violet staining (F), and TUNEL staining (G) in vehicle + middle cerebral artery occlusion (MCAO), retinoic acid (RA) + MCAO, vehicle + sham, and RA + sham animals. Retinoic acid improves neurological behavior deficits (A-C) and histopathological changes (D, F, G) in ischemic brain injury. Intact area were stained dark purple and ischemic area were not stained or stained light purple (D). Retinoic acid alleviated the increase in ischemic area due to MCAO damage. F represents microscopic photos of

the square in photograph D. Arrows indicate damaged neurons with pyknotic nuclei, cytoplasmic vacuolation, and shrunken dendrites (F). Open arrows indicate the TUNEL-positive cells (G). Retinoic acid attenuated the increase in the number of damaged neurons and the number of TUNEL-positive cells caused by MCAO (E and G). Data (neurobehavioral test, $n = 10$; histopathological test, $n = 5$) are shown as the mean ± S.E.M. $^*p < 0.001$, $^{**}p < 0.01$ vs. vehicle + sham animals, #$p < 0.001$, ##$p < 0.01$ vs. vehicle + MCAO animals. Scale bar = 100 μm.

respectively. Furthermore, phospho-Bad levels were 0.36 ± 0.02 and 0.65 ± 0.02 in vehicle + MCAO and retinoic acid + MCAO animals, respectively (Fig 2B). However, there was no significant difference in PDK1 and Akt expression between vehicle- and retinoic acid-treated animals regardless of MCAO damage. The results of immunohistochemical analysis

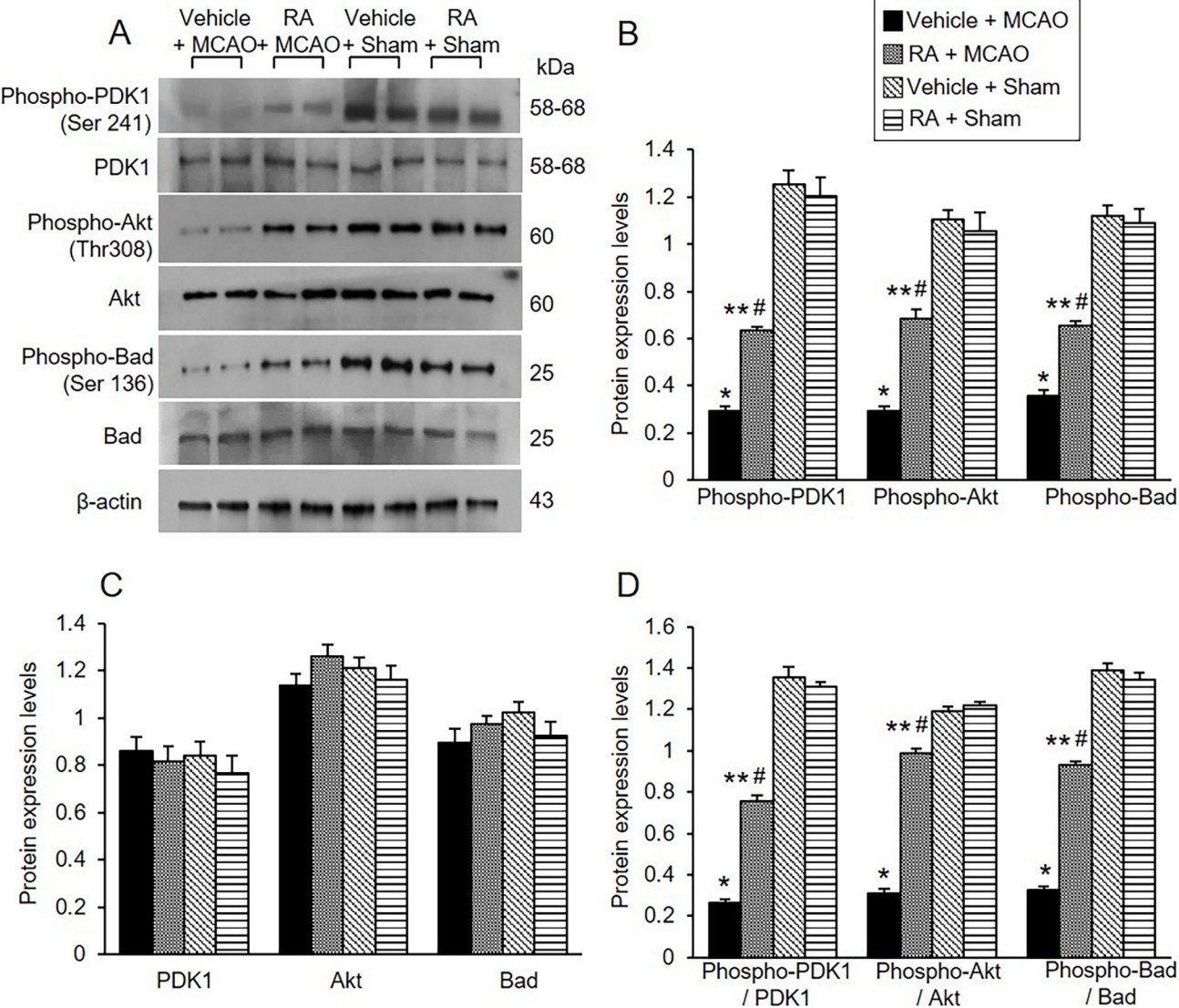

**Fig 2. Retinoic acid alleviates decreases of phospho-Akt and phospho-Bad expression caused by MCAO damage.** Western blot analysis of phospho-PDK1, PDK1, phospho-Akt, Akt, phospho-Bad, and Bad in the cerebral cortex from vehicle + middle cerebral artery occlusion (MCAO), retinoic acid (RA) + MCAO, vehicle + sham, and RA + sham animals. Phospho-PDK1, phospho-Akt, phospho-Bad expressions were decreased in vehicle + MCAO animals, retinoic acid treatment alleviated these decreases. Each lane represents an individual experimental animal. Densitometric analysis is represented as a ratio, proteins intensity to β-actin intensity. Molecular weight (kDa) are depicted at right. Data ($n = 5$) are represented as mean ± S.E.M. $^*p < 0.001$, $^{**}p < 0.01$ vs. vehicle + sham animals, #$p < 0.01$ vs. vehicle + MCAO animals.

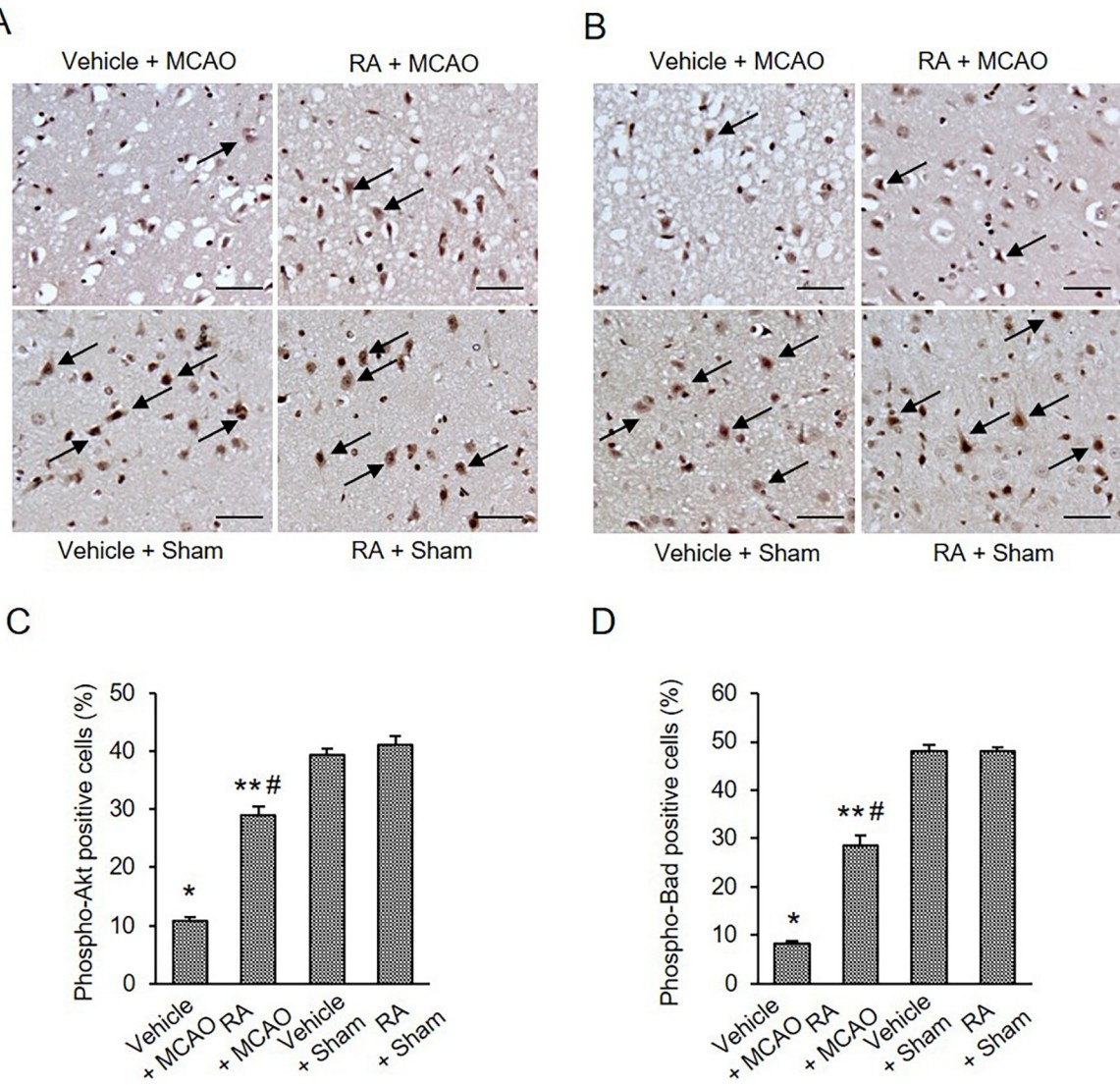

**Fig 3. Retinoic acid attenuates reductions of phospho-Akt and phospho-Bad expression due to MCAO damage.**
Immunohistochemical staining of phospho-Akt (A and C) and phospho-Bad (B and D) in cerebral cortex from vehicle + middle cerebral artery occlusion (MCAO), retinoic acid (RA) + MCAO, vehicle + sham, and RA + sham animals. The positive cells of phospho-Akt and phospho-Bad were stained dark brown. Retinoic acid alleviated the decrease of phospho-Akt and phospho-Bad expression due to MCAO damage. The value of positive cells was expressed as a percentage of the number of positive cells to the number of total cells. Arrows indicate positive cells. Data ($n = 5$) are represented as mean ± S.E.M. *$p < 0.001$, **$p < 0.01$ vs. vehicle + sham animals, #$p < 0.01$ vs. vehicle + MCAO animals. Scale bar = 100 μm.

confirmed the change of phospho-Akt and phospho-Bad expression in all groups (Fig 3). Phospho-Akt and phospho-Bad expression was decreased in vehicle + MCAO animals, retinoic acid treatment mitigated this decrease (Fig 3A and 3B). Positive cells were observed at similar levels between vehicle + sham and retinoic acid + sham animals. The number of phospho-Akt-positive cells was 10.72 ± 0.65% and 28.94 ± 1.58% in vehicle + MCAO and retinoic acid + MCAO animals, respectively (Fig 3C). The number of phospho-Bad-positive cells was 8.27 ± 0.38% in vehicle + MCAO and 28.54 ± 2.17% in retinoic acid + MCAO animals (Fig 3D).

## Regulation of phospho-Bad and 14-3-3 interaction by retinoic acid in MCAO animals

We examined the expression of 14-3-3 in the cerebral cortex of all groups. MCAO damage slightly reduced 14-3-3 expression, and retinoic acid treatment attenuated this reduction (Fig 4A). The 14-3-3 levels were 0.87 ± 0.03 and 0.97 ± 0.04 in vehicle + MCAO and retinoic acid + MCAO animals, respectively (Fig 4B). Immunoprecipitation was performed to determine the interaction between phospho-Bad and 14-3-3 (Fig 4C). Binding levels of these proteins were reduced in vehicle-treated animals with MCAO damage compared with sham animals. These reductions were alleviated in retinoic acid-treated animals with MCAO. Interaction levels were 0.32 ± 0.02 in vehicle + MCAO animals and 0.56 ± 0.02 in retinoic acid + MCAO animals (Fig 4D).

## Regulation of Bcl-2 family protein interactions by retinoic acid in MCAO animals

We also observed Bcl-2/Bad and Bcl-xL/Bad binding levels. MCAO damage increased Bcl-2/Bad and Bcl-xL/Bad binding, and retinoic acid treatment attenuated these increases (Fig 5A). Bcl-2/Bad binding levels were 1.27 ± 0.06 in vehicle + MCAO animals and 1.10 ± 0.04 in retinoic acid + MCAO animals (Fig 5C). Furthermore, Bcl-xL/Bad binding levels were 0.97 ± 0.03

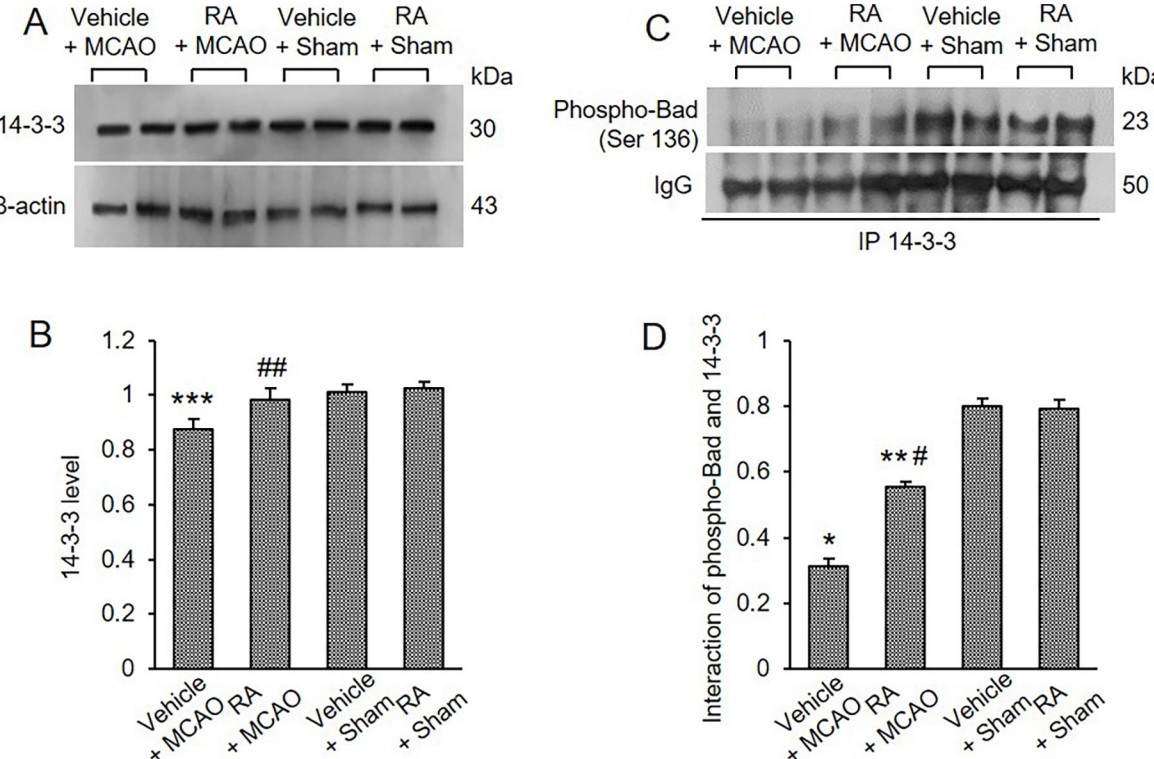

**Fig 4. Retinoic acid alleviates decrease of phospho-Bad and 14-3-3 interaction caused by MCAO damage.** Western blot analysis of 14-3-3 (A and B) and immunoprecipitation analysis of phospho-Bad and 14-3-3 binding (C and D) in the cerebral cortex from vehicle + middle cerebral artery occlusion (MCAO), retinoic acid (RA) + MCAO, vehicle + sham, and RA + sham animals. Retinoic acid attenuated the decrease of phospho-Bad and 14-3-3 interaction caused by MCAO damage. Each lane represents an individual experimental animal. Densitometric analysis is represented as a ratio, proteins intensity to β-actin (B) or IgG (D) intensity. Molecular weights (kDa) are depicted at right. Data ($n = 5$) are represented as mean ± S.E.M. $^*p < 0.001$, $^{**}p < 0.01$, $^{***}p < 0.05$ vs. vehicle + sham animals, $^\#p < 0.01$, $^{\#\#}p < 0.05$ vs. vehicle + MCAO animals.

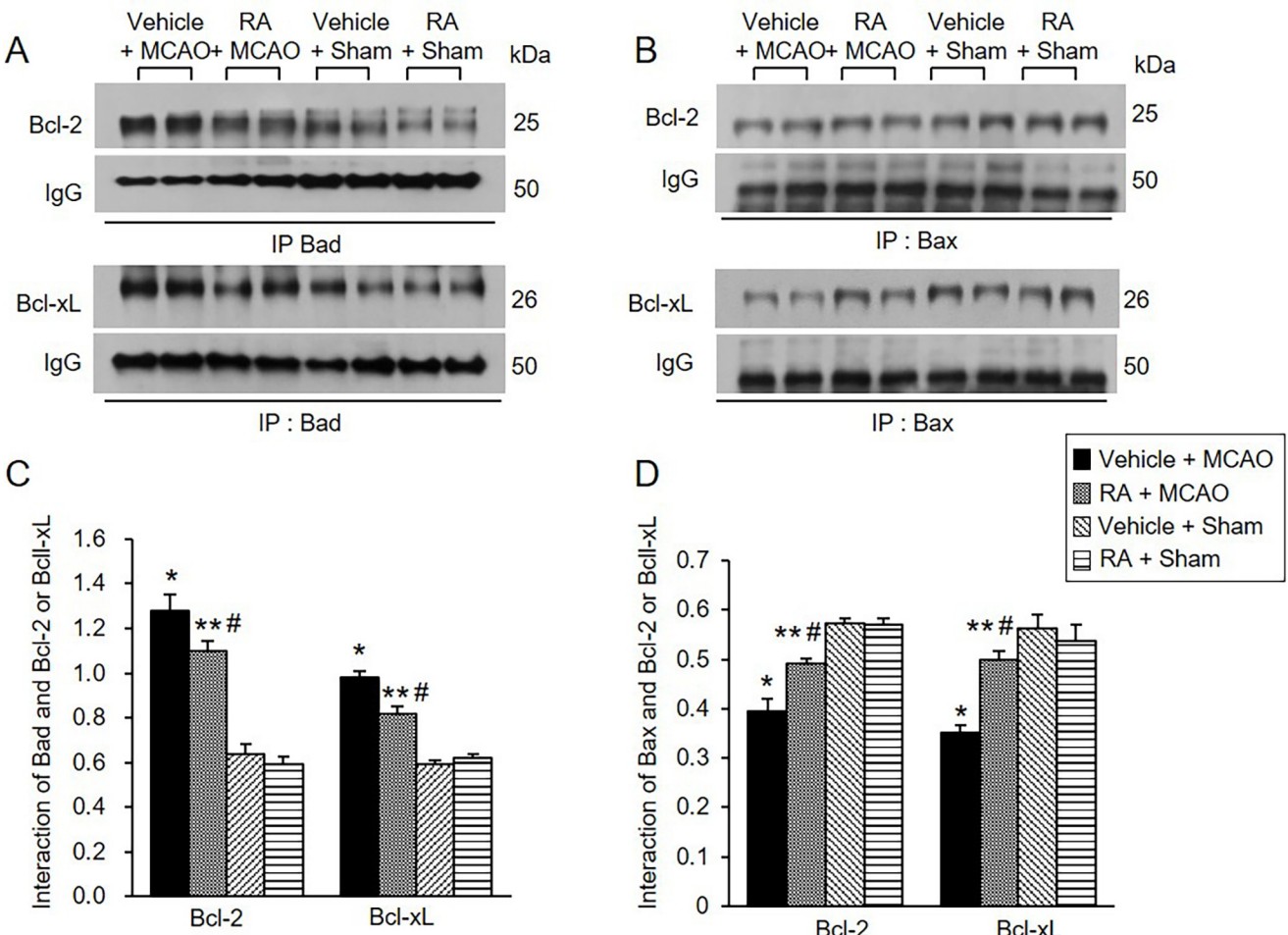

**Fig 5. Retinoic acid regulates Bcl-2 family protein interactions in MCAO animals.** Immunoprecipitation analysis of Bcl-2 and Bcl-xL to Bad (A and C) or Bax (B and D) in the cerebral cortex from vehicle + middle cerebral artery occlusion (MCAO), retinoic acid (RA) + MCAO, vehicle + sham, and RA + sham animals. Retinoic acid alleviated the increase of Bcl-2/Bad and Bcl-xL/Bad binding and the decrease of Bcl/Bax and Bcl-xL and Bcl-xL/Bax caused by MCAO damage. Each lane represents an individual experimental animal. Densitometric analysis is represented as a ratio of proteins intensity to IgG intensity. Molecular weights (kDa) are depicted at right. Data ($n = 5$) are represented as mean ± S.E.M. *$p < 0.001$, **$p < 0.01$ vs. vehicle + sham animals, #$p < 0.01$ vs. vehicle + MCAO animals.

and 0.82 ± 0.04 in vehicle + MCAO and retinoic acid + MCAO animals, respectively (Fig 5C). In addition, Bcl-2/Bax and Bcl-xL/Bax binding levels were evaluated and found to be decreased in vehicle-treated animals with MCAO. These decreases were attenuated in retinoic acid-treated animals with MCAO (Fig 5B). Bcl-2/Bax binding levels were 0.39 ± 0.02 in vehicle + MCAO animals and 0.49 ± 0.01 in retinoic acid + MCAO animals (Fig 5D). Bcl-xL/Bax binding levels were 0.35 ± 0.02 and 0.50 ± 0.02 in vehicle + MCAO and retinoic acid + MCAO animals, respectively (Fig 5D).

## Modulation of caspase-3 expression by retinoic acid in MCAO animals

We also confirmed the expression of caspase-3 protein, a representative marker of apoptosis. Retinoic acid alleviated the increase of caspase-3 and cleaved caspase-3 expression caused by MCAO (Fig 6A). Caspase-3 levels were 1.31 ± 0.03 in vehicle + MCAO animals and 0.81 ± 0.06 in retinoic acid + MCAO animals (Fig 6B). Cleaved-caspase-3 levels were

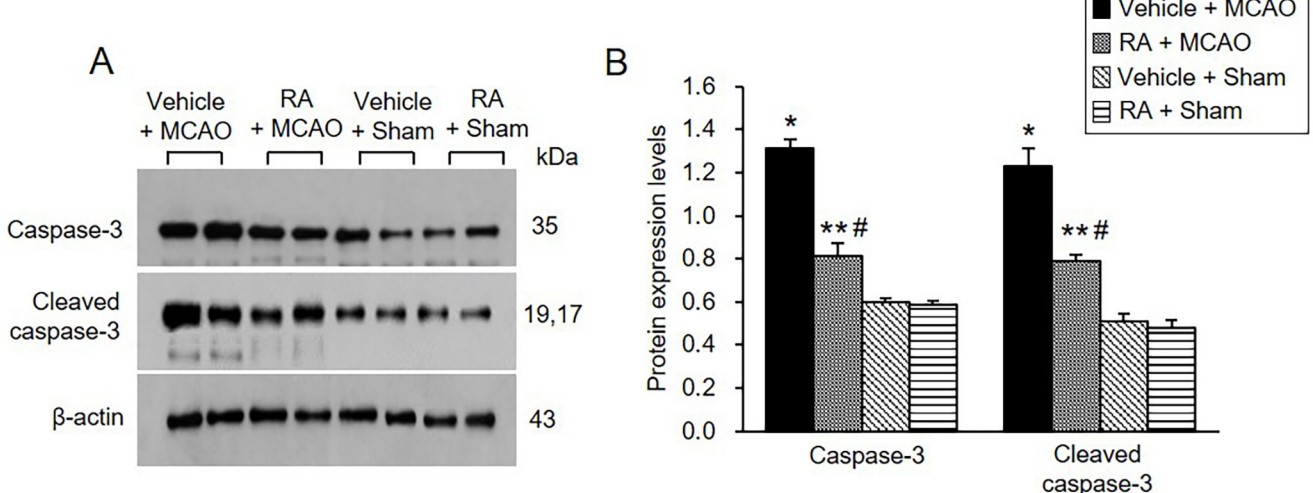

**Fig 6. Retinoic acid attenuates increases of caspase-3 and cleaved caspase-3 expression caused by MCAO damage.** Western blot analysis of caspase-3 and cleaved caspase-3 in the cerebral cortex from vehicle + middle cerebral artery occlusion (MCAO), retinoic acid (RA) + MCAO, vehicle + sham, and RA + sham animals. Retinoic acid alleviated the increase of these proteins due to MCAO. Each lane represents an individual experimental animal. Densitometric analysis is represented as a ratio of proteins intensity to β-actin intensity. Molecular weights (kDa) are depicted at right. Data ($n = 5$) are represented as mean ± S.E.M. $*p < 0.001$, $**p < 0.01$ vs. vehicle + sham animals, $\#p < 0.01$ vs. vehicle + MCAO animals.

1.23 ± 0.07 and 0.79 ± 0.03 in vehicle + MCAO and retinoic acid + MCAO animals, respectively (Fig 6B).

## Discussion

This study supported the neuroprotective effects of retinoic acid through various neurological behavior tests including neurological deficits scoring test, corner test, and grip strength test. Surgical MCAO was performed in the right brain of rats. A reduction of grip strength was observed in the left forelimb. These results showed neurological behavioral deficits in the contralateral side of the damaged brain. Retinoic acid mitigated the neurological disorder caused by MCAO and exerted neuroprotective effects. Furthermore, the morphological changes were confirmed in cresyl violet-stained tissues. Retinoic acid treatment decreased infarct areas caused by MCAO damage and reduced the number of damaged cells. We recently reported that retinoic acid alleviates the increases in brain edema and infarction volume due to MCAO damage [41]. The present study also confirmed that retinoic acid attenuates the histopathological changes and increase in the number of TUNEL-positive cells in the cerebral cortex with MCAO. Stroke is a complex disease involving a variety of mechanisms. The PI3K-Akt pathway is a representative pathway for neuronal survival and protection. PI3K-Akt is a key mediator of cerebral ischemic stroke [37]. The present study focused on the neuroprotective effects of retinoic acid by modulating the PI3K-Akt signaling pathway. This study additionally elucidated that retinoic acid performs neuroprotective functions through activation of the PI3K-Akt-Bad signaling pathway and regulation of Bcl-2 family protein interactions in a stroke animal model.

The PI3K-Akt signaling pathway regulates the survival and apoptosis of neurons [46, 47]. Activated Akt phosphorylates Bad and prevents the apoptotic function of Bad, leading to survival of neurons [48, 49]. However, in hypoxia and ischemia states, inactivated Akt dephosphorylates Bad and induces apoptotic cell death [50]. We previously showed that the PI3K/Akt signaling pathway was associated with neuroprotective mechanisms of various substances in

ischemic stroke models [51–53]. We also reported that these substances exert neuroprotective effects by regulating the expression of Akt downstream targets such as Bad, FKHR, and GSK-3β [54]. Therefore, we postulate that activation of Akt plays an important role in protecting neurons from stroke damage. Furthermore, retinoic acid activates the Akt pathway and protects cells against proteasome inhibition-associated cell death [33]. In the present study, retinoic acid alleviated the decrease in phospho-Akt as well as the reduction in phospho-PDK1 and phospho-Bad expression due to MCAO damage. PDK1 and Bad are upstream and downstream target proteins of Akt, respectively. PDK1 and Akt expression was maintained constant regardless of MCAO damage. Thus, the results provide evidence that phosphorylation of these proteins is important for performing neuroprotective effects against ischemic damage rather than changes in the total protein expression levels. In addition, immunohistochemical staining confirmed the changes in phospho-Akt and phospho-Bad expression were confirmed based on. The results were consistent with the results of Western blot analysis. MCAO decreased the number of positive cells in phospho-Akt and phospho-Bad, and retinoic acid attenuated this decrease. Bad is a representative Akt downstream protein and a proapoptotic protein. Previous studies have demonstrated the efficacy of retinoic acid in various diseases [55, 56]. Especially in the brain, retinoic acid has neuroprotective effects in neurodegenerative diseases such as amyotrophic lateral sclerosis, Parkinson's and ischemic stroke. In this study, we focused on the specific protective mechanism of retinoic acid on cerebral ischemia [57–59]. The PI3K-Akt pathway has shown therapeutic and preventive effects in various neurological disorders including stroke, Alzheimer's disease, and Parkinson's disease [60, 61]. Therefore, it is hypothesized that the regulation of the PI3K-Akt pathway by retinoic acid treatment may also be involved in the prevention and treatment of stroke. We previously showed a decrease in phospho-Bad due to MCAO damage and further described the mitigation of phospho-Bad decrease caused by retinoic acid. The changes in phospho-Bad expression due to retinoic acid treatment in MCAO indicate the importance of phospho-Bad in neuroprotective effects of retinoic acid. The maintenance of phospho-Akt and phospho-Bad plays a crucial role in cell survival. Thus, the results show that the PI3K/Akt signaling pathway contributes to the neuroprotective mechanism of retinoic acid in cerebral ischemia.

The interaction between phospho-Bad and 14-3-3 is important for cell survival. The binding of these proteins inhibits interaction with Bad and Bcl-xL or Bcl-2, induces Bcl-2/Bax or Bcl-xL/Bax complexes, inactivates pro-apoptotic function of Bax, and continuously prevents caspase cascade activation and apoptosis [20, 21, 24, 62]. The results of immunoprecipitation analysis showed a decrease in phospho-Bad and 14-3-3 binding in cerebral cortex with MCAO damage, which was attenuated in the presence of retinoic acid. The findings demonstrate that retinoic acid regulates phospho-Bad and 14-3-3 binding in rats with MCAO damage. Thus, the results provide evidence that retinoic acid regulates phospho-Bad and 14-3-3 binding and preserves neurons from ischemic damage. The binding of Bcl-2 and Bcl-xL to Bad or Bax in the cerebral cortex of MCAO animals was investigated. Bcl-2/Bad and Bcl-xL/Bad binding increased in MCAO-damaged rats, and retinoic acid prevented this increase. However, Bcl-2/Bax and Bcl-xL/Bax binding in rats with MCAO damage was reduced, and retinoic acid attenuated this decrease. Phosphorylated Bad prevented Bcl-2/Bad and Bcl-xL/Bad binding and inhibited the pro-apoptotic function of Bad. In addition, phosphorylated Bad induced Bcl-2/Bax and Bcl-xL/Bax binding and prevented pro-apoptotic activity of Bax. Our findings showed that retinoic acid alleviates the increase of Bcl-2/Bad and Bcl-xL/Bad binding and the decrease of Bcl-2/Bax and Bcl-xL/Bax binding due to MCAO damage. Retinoic acid regulates the phosphorylation of Bad and continuously modulates the interaction of Bcl-2 family proteins, eventually inhibiting apoptosis. We previously reported a decrease in Bcl-2 and an increase in Bax in MCAO damage, and retinoic acid attenuated those changes [41]. Furthermore, retinoic acid alleviated the increase in Bcl-2/Bax ratio caused by MCAO damage. The results show that

retinoic acid exerts neuroprotective effects by regulating Bcl-2 family protein expression. In addition, we observed change in caspase-3 and cleaved caspase-3 expression, indicators of apoptosis. Retinoic acid significantly attenuated increases of caspase-3 and cleaved caspase-3 expression caused by MCAO. Our previous study showed that retinoic acid alleviates the increase of caspase-9, cleaved caspase-9, PARP, and cleaved-PARP in MCAO animals [41]. These results provide additional information confirming the current findings that retinoic acid prevents apoptosis from cerebral ischemic damage and protects brain tissue. Retinoic acid mitigated the decrease in phospho-Bad expression. It also alleviated the increase in Bcl-2/Bad and Bcl-xL/Bad binding and the decrease of Bcl-2/Bax and Bcl-xL/Bax in MCAO animals. The results indicate that retinoic acid inhibits the pro-apoptotic function of Bad and Bax. Finally, we confirmed that retinoic acid prevents the increase in caspase-3 and cleaved caspase-3 activity, thereby inhibiting apoptosis and protecting neurons from ischemic damage. The neuroprotective effect of retinoic acid on MCAO damage is due to Akt signaling pathway regulation and Bcl-2 family protein regulation (Fig 7). Retinoic acid exerts neuroprotective effects

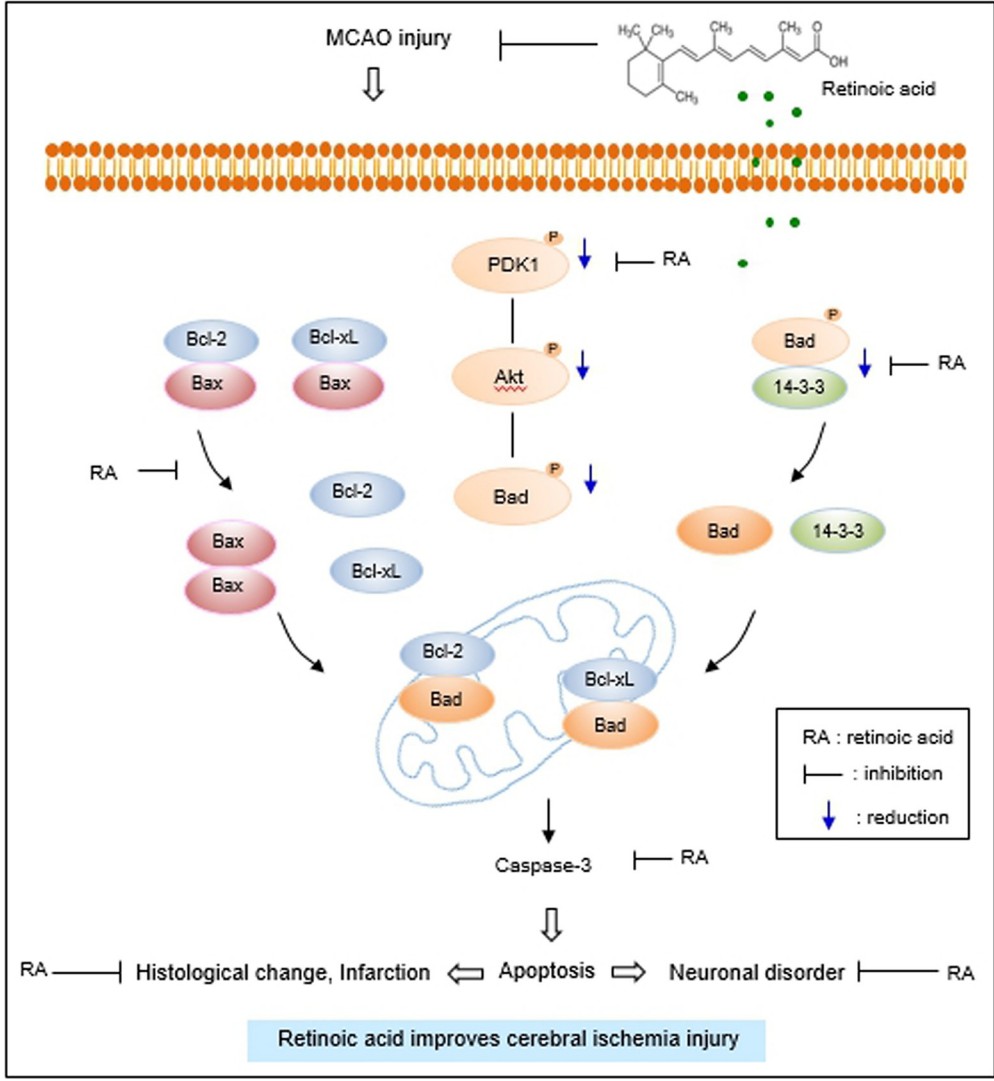

**Fig 7. The neuroprotective mechanism of retinoic acid against MCAO damage.**

through various protective mechanisms during brain injury. However, it is difficult to discuss all protective mechanism of retinoic acid in this study. We focused on the regulatory mechanism of Akt and its downstream target Bad by retinoic acid in cerebral ischemia. We clearly showed that retinoic acid is involved in neuroprotection through the regulation of Akt and Bad in ischemic brain injury. We think that various protective mechanisms of retinoic acid including upstream and downstream target of Akt should be researched in the future. We also need to find more factors that are regulated by retinoic acid in the ischemic stroke. Further studies should be performed that provide therapeutic effects of retinoic acid in stoke model.

## Conclusions

Taken together, our results demonstrated that retinoic acid treatment induces the phosphorylation of Akt and Bad, regulates the interaction with Bcl-2 family proteins, and modulates the expression of caspase-3 in stroke animal models (S4–S11 Files). Retinoic acid inhibits the process of apoptosis, promotes neuronal survival, and ultimately exerts neuroprotective effects in ischemic stroke (S1–S3 Files). In conclusion, retinoic acid provides neuroprotective effects through mitigation of Akt and Bad phosphorylation and modulation of Bcl-2 family protein interactions in ischemic stroke.

## Supporting information

**S1 File. Gross photographs of cresyl violet staining.** This is full images of Fig 1D.
(PDF)

**S2 File. Microscopic photographs of cresyl violet staining.** Full images of Fig 1F.
(PDF)

**S3 File. TUNEL staining.** Full images of Fig 1G.
(PDF)

**S4 File. Western blot analysis of phospho-PDK1, PDK1, phospho-Akt, Akt, phospho-Bad, and Bad in the cerebral cortex.** Full images of Fig 2A.
(PDF)

**S5 File. Immunohistochemical staining of phospho-Akt.** Full images of Fig 3A.
(PDF)

**S6 File. Immunohistochemical staining of phospho-Bad.** Full images of Fig 3B.
(PDF)

**S7 File. Western blot analysis of 14-3-3.** Full images of Fig 4A.
(PDF)

**S8 File. Immunoprecipitation analysis of phospho-Bad and 14-3-3 binding.** Full images of Fig 4C.
(PDF)

**S9 File. Immunoprecipitation analysis of Bcl-2 and Bcl-xL to Bad.** Full images of Fig 5A.
(PDF)

**S10 File. Immunoprecipitation analysis of Bcl-2 and Bcl-xL to Bax.** Full images of Fig 5B.
(PDF)

**S11 File. Western blot analysis of caspase-3 and cleaved caspase-3.** Full images of Fig 6A.
(PDF)

## Author Contributions

**Conceptualization:** Phil-Ok Koh.

**Data curation:** Ju-Bin Kang.

**Funding acquisition:** Phil-Ok Koh.

**Methodology:** Ju-Bin Kang.

**Software:** Ju-Bin Kang.

**Supervision:** Phil-Ok Koh.

**Visualization:** Ju-Bin Kang.

**Writing – original draft:** Ju-Bin Kang, Phil-Ok Koh.

**Writing – review & editing:** Phil-Ok Koh.

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
