## [Decision Letter · Decision Letter 0]

28 Nov 2023

PONE-D-23-19266Retinoic acid alleviates the reduction of Akt and Bad phosphorylation and regulates Bcl-2 family protein interactions in animal models of ischemic strokePLOS ONE

Dear Dr. Koh,

Thank you for submitting your manuscript to PLOS ONE. After careful consideration, we feel that it has merit but does not fully meet PLOS ONE’s publication criteria as it currently stands. Therefore, we invite you to submit a revised version of the manuscript that addresses the points raised during the review process.

We look forward to receiving your revised manuscript.

Kind regards,

Ahmed E. Abdel Moneim

Academic Editor

PLOS ONE

Journal Requirements:

"The funders had no role in study design, data collection and analysis, decision to publish, or preparation of the manuscript"

Reviewers' comments:

Reviewer's Responses to Questions

**Comments to the Author**

1. Is the manuscript technically sound, and do the data support the conclusions?

Reviewer #1: Partly

Reviewer #2: Yes

Reviewer #3: Yes

2. Has the statistical analysis been performed appropriately and rigorously? 

Reviewer #1: Yes

Reviewer #2: Yes

Reviewer #3: Yes

3. Have the authors made all data underlying the findings in their manuscript fully available?

Reviewer #1: Yes

Reviewer #2: Yes

Reviewer #3: Yes

4. Is the manuscript presented in an intelligible fashion and written in standard English?

Reviewer #1: Yes

Reviewer #2: Yes

Reviewer #3: Yes

5. Review Comments to the Author

Reviewer #1: Major Revisions

The authors showed that retinoic acid pretreatment can alleviate stroke induced by middle cerebral artery occlusion. They showed this by neurological scoring and other tests with histopathological changes and western blot analysis for protein expressions.

1. The author fails to show a direct mechanism in which retinoic acid is exerting the effect observed.

2. The signaling pathway observed is not the only major pathway involved in stroke, other pathways have been implicated as well. Moreso, retinoic acid not only activates Akt/PI3K pathway alone. The changes in other stroke pathways in relation to retinoic acid should also be investigated.

Minor revisions

1. Duration of the occlusion not mentioned.

2. Interval between the surgery and testing if 24 hours is short.

3. How did the authors avoid bias in their scoring?

4. The number of training sessions/day and interval of testing for corner test is missing.

5. Similarly, the interval of gripping test is missing.

6. Sampling of brain sections for histopathological analysis was not adequately mentioned. Were the sections selected randomly or a systemic approach?

7. There are some typos and grammar that need to be revised.

Other

Abstract

Line 30 – ‘and cerebral cortical tissues’ hanging statement.

Line 31 – analysis wrongly spelled.

Line 38 – ‘it alleviated………” need revision.

Line 39 – tautology

Introduction

Line 48-49 – need revision.

Methods

Line 111 – “Animals were performed neuronal behavioral tests” animals were assessed on neurobehavioral tests.

Line 123 – cm3 remove 3.

Line 137 – overnight wash with tap water – are you sure about this?

Duration of dehydrating and rehydrating of slides

Line 166 – “cleared” not “cleaned”.

Line 175 – “in ice” replace with “on ice”.

Line 287 – remove “continuously”.

Discussion

Line 316 – remove “strongly”.

Reviewer #2: In this research, the author provided the data about the neuroprotective effects of retinoic acid through various neurological behavior tests including neurological deficits scoring test, corner test, and grip strength test. And explained its mechanism that retinoic acid prevented apoptosis against cerebral ischemia through phosphorylation of Akt and Bad, maintenance of phospho-Bad and 14-3-3 binding, and regulation of Bcl-2

family protein interactions. Most of the results seem likely to be valid as experimental observations.

Minor points:

Please detect the phosphorylation status of brain tissue after MCAO injury.

Reviewer #3: 1) the references cited in the past three years are not sufficient and need to be updated. 2) The statistical significance in the annotations is not consistent with that in the figure and needs to be unified. 3) When exploring the protective mechanism of Retinoic acid, only the phenomenon has been observed in this study. Although the results obtained were encouraging, there was no validation through the reverse way. Therefore, the results of this study are not so reliable and powerful. Therefore, it is recommended not to use words such as “demonstrated” when describing the results of this study.

6. PLOS authors have the option to publish the peer review history of their article (what does this mean?). If published, this will include your full peer review and any attached files.

Reviewer #1: No

Reviewer #2: No

Reviewer #3: No

---

## [Author Response · Author response to Decision Letter 0]

28 Dec 2023

Reviewer #1: Major Revisions

The authors showed that retinoic acid pretreatment can alleviate stroke induced by middle cerebral artery occlusion. They showed this by neurological scoring and other tests with histopathological changes and western blot analysis for protein expressions.

1. The author fails to show a direct mechanism in which retinoic acid is exerting the effect observed.

Response:

We appreciate for your kind comments. We tried to answer your questions very carefully.

Retinoic acid has several properties such as anti-oxidation (Ahlemeyer., 2001), anti-inflammation (Oliveira et al., 2018), anti-cancer (Hunsu et at., 2021). Especially in the brain, retinoic acid has neuroprotective effects in neurodegenerative diseases such as amyotrophic lateral sclerosis (Zhu et al., 2020), Parkinson's disease (Esteves et al., 2015) and ischemic stroke (Cai et al., 2019). In this study, we focused on the specific protective mechanism of retinoic acid on cerebral ischemia. The signal pathways involved in stroke is very diverse and complex. The regulatory pathways of retinoic acid in ischemic strokes are also diverse and complex. Among these multiple pathways, we focused on the PI3K/Akt pathway, which is a signal transduction pathway that promotes survival and growth in response to extracellular signals. Akt pathway regulates glucose metabolism, protein synthesis, mitochondrial metabolism, lipid metabolism, angiogenesis, autophagy, proliferation, and cell growth (Avan et al., 2016). It is a representative pathway related to cell survival. The regulation of Akt pathway by retinoic acid in stroke has not been reported much. Akt inhibits apoptosis through phosphorylation of proapoptotic proteins such as Bad and forkhead transcription factor (FKHR). However, dephosphorylated Bad forms heterodimers with Bcl-2 and Bcl-xL to inactivate them, allowing apoptosis. We showed that retinoic acid treatment alleviates MCAO-induced decreases in phospho-PDK1, phospho-Akt and phospho-Bad expression. Retinoic acid attenuated decrease in interaction between phospho-Bad and 14-3-3 caused by MCAO damage. Furthermore, retinoic acid mitigated the increase in Bcl-2/Bad and Bcl-xL/Bad binding levels and the reduction in Bcl-2/Bax and Bcl-xL/Bax binding levels caused by MCAO damage. Retinoic acid alleviated MCAO-induced increase of caspase-3 and cleaved caspase-3 expression. We demonstrate that retinoic acid prevented apoptosis against cerebral ischemia through phosphorylation of Akt and Bad, maintenance of phospho-Bad and 14-3-3 binding, and regulation of Bcl-2 family protein interactions. Retinoic acid exerts neuroprotective effects through various protective mechanisms during brain injury. However, it is difficult to discuss all protective mechanism of retinoic acid in this study. Among these various mechanisms, we focused on the PI3K/Akt signal pathway, which is a representative survival pathway. Previous studies demonstrated that retinoic acid activated the Akt signaling pathway and promoted cell survival in various tissues. However, data on the neuroprotective effects of retinoic acid on the PI3K-Akt signaling pathway in cerebral ischemia are limited. It has not been previously reported whether retinoic acid regulates the expression of phospho-Bad and the interaction between phospho-Bad and 14-3-3 in a stroke animal model. Therefore, we focused on the regulatory mechanism of Akt and its downstream target Bad by retinoic acid in cerebral ischemia. We regret that this study did not show various neuroprotective mechanism of retinoic acid in stroke animal model. However, we clearly showed that retinoic acid is involved in neuroprotection through the regulation of Akt and its downstream targets Bad in ischemic brain injury. We suggest that various protective mechanisms of retinoic acid should be researched. Thus, we plan to study other mechanisms in future research with reference to your kind comments.

References

Ahlemeyer B, Bauerbach E, Plath M, Steuber M, Heers C, Tegtmeier F, et al. Retinoic acid reduces apoptosis and oxidative stress by preservation of SOD protein level. Free Radic Biol Med. 2001;30(10):1067-77. 

Oliveira LM, Teixeira FME, Sato MN. Impact of retinoic acid on immune cells and inflammatory diseases. Mediators Inflamm. 2018;2018:3067126.

Hunsu VO, Facey COB, Fields JZ, Boman BM. Retinoids as chemo-preventive and molecular-targeted anti-cancer therapies. International Journal of Molecular Sciences. 2021; 22(14):7731. 

Zhu Y, Liu Y, Yang F, Chen W, Jiang J, He P, et al. All-trans retinoic acid exerts neuroprotective effects in amyotrophic lateral sclerosis-like Tg (SOD1*G93A)1Gur mice. Mol Neurobiol. 2020;57(8):3603-3615.

Esteves M, Cristóvão AC, Saraiva T, Rocha SM, Baltazar G, Ferreira L, et al. Retinoic acid-loaded polymeric nanoparticles induce neuroprotection in a mouse model for Parkinson's disease. Front Aging Neurosci. 2015;7:20.

Cai W, Wang J, Hu M, Chen X, Lu Z, Bellanti JA, et al. All trans-retinoic acid protects against acute ischemic stroke by modulating neutrophil functions through STAT1 signaling. J Neuroinflammation. 2019;16(1):175.

Avan A, Narayan R, Giovannetti E, Peters GJ. Role of Akt signaling in resistance to DNA-targeted therapy. World J Clin Oncol. 2016;7(5):352-369.

We added a sentence to the discussion section to make it easier to understand.

Retinoic acid exerts neuroprotective effects through various protective mechanisms during brain injury. However, it is difficult to discuss all protective mechanism of retinoic acid in this study. We focused on the regulatory mechanism of Akt and its downstream target Bad by retinoic acid in cerebral ischemia. We clearly showed that retinoic acid is involved in neuroprotection through the regulation of Akt and Bad in ischemic brain injury. We suggest that various protective mechanisms of retinoic acid should be researched. 

2. The signaling pathway observed is not the only major pathway involved in stroke, other pathways have been implicated as well. Moreso, retinoic acid not only activates Akt/PI3K pathway alone. The changes in other stroke pathways in relation to retinoic acid should also be investigated.

Response: 

We appreciate for your kind comments

Ischemic stroke is caused primarily by an interruption of cerebral blood flow, causes severe nerve damage, and is one of the leading causes of death and disability worldwide. It induces cell excitotoxicity, mitochondrial dysfunction, neuroinflammation, blood-brain barrier damage, and apoptotic processes. It is also known that the mechanism of causing ischemic stroke is very complicated. Moreover, the signaling pathway of ischemic stroke is very diverse and complex. The typical signal pathway involved in stroke is as follows; phosphatidylinositol 3-kinase (PI3K)-Akt signaling pathway, phosphatase and tensin homolog (PTEN) signaling pathway, death-associated protein kinase 1 (DAPK1) signaling pathway, postsynaptic density protein-95 (PSD95)/neuronal nitric oxide synthase (nNOS) signaling pathways, hypoxia-inducible factor (HIF) signaling pathway, nuclear factor E2-related factor 2 (Nrf2) signaling pathway, casein kinase 2 (CK2) signaling pathway, mTOR-related signaling pathways, and p53-mediated apoptotic pathway (Jo et al., 2012, Zhang et al., 2013, Shamloo et al., 2005, Guo et al., 2009, Dinkova-Kostova and Abramov, 2015, Bastian et al., 2019, Perez-Alvarez et al., 2018, Leker et al., 2004). As mentioned above, stroke is associated with various signaling pathways, but it is difficult to discuss all pathways in this study. Among these various signaling pathways, we focused on the PI3K/Akt signal pathway, which is a representative survival pathway. We investigated the change of Akt and its downstream target Bad by retinoic acid in cerebral ischemia.

Reference

Jo H, Mondal S, Tan D, Nagata E, Takizawa S, Sharma AK, et al. Small molecule-induced cytosolic activation of protein kinase Akt rescues ischemia-elicited neuronal death. Proc Natl Acad Sci USA. 2012;109: 10581-10586.

Zhang S, Taghibiglou C, Girling K, Dong Z, Lin SZ, Lee W, et al. Critical role of increased PTEN nuclear translocation in excitotoxic and ischemic neuronal injuries. J Neurosci. 2013;33: 7997-8008.

Shamloo M, Soriano L, Wieloch T, Nikolich K, Urfer R, Oksenberg D. Death-associated protein kinase is activated by dephosphorylation in response to cerebral ischemia. J Biol Chem. 2005;280: 42290-4229.

Guo S, Miyake M, Liu KJ, Shi H. Specific inhibition of hypoxia inducible factor 1 exaggerates cell injury induced by in vitro ischemia through deteriorating cellular redox environment. J Neurochem. 209;108: 1309-1321.

Dinkova-Kostova AT, Abramov AY. The emerging role of Nrf2 in mitochondrial function. Free Radic Biol Med. 2015;88: 179-188.

Bastian C, Quinn J, Tripathi A, Aquila D, McCray A, Dutta R, et al. CK2 inhibition confers functional protection to young and aging axons against ischemia by differentially regulating the CDK5 and AKT signaling pathways. Neurobiol Dis. 2019;126: 47-61.

Perez-Alvarez MJ, Villa Gonzalez M, Benito-Cuesta I, Wandosell FG. Role of mTORC1 controlling proteostasis after brain ischemia. Front Neurosci. 2018;12: 60.

Leker RR, Aharonowiz M, Greig NH, Ovadia H. The role of p53-induced apoptosis in cerebral ischemia: effects of the p53 inhibitor pifithrin alpha. Exp Neurol. 2004;187: 478-486.

In addition, retinoic acid plays a variety of roles including anti-oxidation, anti-inflammation, and anti-cancer., Retinoic acid inhibited cell death through regulation of apoptosis-related proteins (Kang et al., 2021). It regulates calcium concentration through the regulation of calcium binding proteins and inhibits apoptosis in response to brain ischemic injury (Kang et al., 2023). It also modulated MAP kinase pathway and caspase cascade. In previous studies, retinoic acid activated the Akt signaling pathway and promoted cell survival in various tissues. However, data on the neuroprotective effects of retinoic acid on the PI3K-Akt signaling pathway in cerebral ischemia are limited. It has not been previously reported whether retinoic acid regulates the expression of phospho-Bad and the interaction between phospho-Bad and 14-3-3 in a stroke animal model. Therefore, this study was designed to investigate the neuroprotective effects of retinoic acid on cerebral ischemia and the regulation of phospho-Akt and phospho-Bad by retinoic acid. In addition, we examined whether retinoic acid inhibits apoptosis and protects brain tissues from cerebral ischemia by controlling the binding of phospho-Bad with 14-3-3 and binding of Bcl-2 and Bcl-xL to Bad or Bax. Therefore, we focused on the regulatory mechanism of AKT and its downstream target Bad by retinoic acid in cerebral ischemia. 

We added sentences to introduction section to make it easier to understand.

Ischemic stroke induces cell excitotoxicity, mitochondrial dysfunction, blood-brain barrier damage, neuroinflammation, and apoptotic processes. It is also known that the mechanism of causing ischemic stroke is very complicated. Moreover, the signaling pathway of ischemic stroke is very diverse and complex. The signal pathways involved in stroke are PI3K/Akt signaling pathway, phosphatase and tensin homolog signaling pathway, death-associated protein kinase 1 signaling pathway, neuronal nitric oxide synthase signaling pathways, hypoxia-inducible factor signaling pathway, nuclear factor E2-related factor 2 signaling pathway, casein kinase 2 signaling pathway, mTOR-related signaling pathways, and p53-mediated apoptotic pathway [23-30]. As mentioned above, stroke is associated with various signaling pathways, but it is difficult to discuss all pathways in this study. Among these various signaling pathways, we focused on the PI3K/Akt signal pathway, which is a representative survival pathway

We added references to reference list.

23. Jo H, Mondal S, Tan D, Nagata E, Takizawa S, Sharma AK, et al. Small molecule-induced cytosolic activation of protein kinase Akt rescues ischemia-elicited neuronal death. Proc Natl Acad Sci USA. 2012;109: 10581-10586.

24. Zhang S, Taghibiglou C, Girling K, Dong Z, Lin SZ, Lee W, et al. Critical role of increased PTEN nuclear translocation in excitotoxic and ischemic neuronal injuries. J Neurosci. 2013;33: 7997-8008.

25. Shamloo M, Soriano L, Wieloch T, Nikolich K, Urfer R, Oksenberg D. Death-associated protein kinase is activated by dephosphorylation in response to cerebral ischemia. J Biol Chem. 2005;280: 42290-4229.

26. Guo S, Miyake M, Liu KJ, Shi H. Specific inhibition of hypoxia inducible factor 1 exaggerates cell injury induced by in vitro ischemia through deteriorating cellular redox environment. J Neurochem. 209;108: 1309-1321.

27. Dinkova-Kostova AT, Abramov AY. The emerging role of Nrf2 in mitochondrial function. Free Radic Biol Med. 2015;88: 179-188.

28. Bastian C, Quinn J, Tripathi A, Aquila D, McCray A, Dutta R, et al. CK2 inhibition confers functional protection to young and aging axons against ischemia by differentially regulating the CDK5 and AKT signaling pathways. Neurobiol Dis. 2019;126: 47-61.

29. Perez-Alvarez MJ, Villa Gonzalez M, Benito-Cuesta I, Wandosell FG. Role of mTORC1 controlling proteostasis after brain ischemia. Front Neurosci. 2018;12: 60.

30. Leker RR, Aharonowiz M, Greig NH, Ovadia H. The role of p53-induced apoptosis in cerebral ischemia: effects of the p53 inhibitor pifithrin alpha. Exp Neurol. 2004;187: 478-486.

Minor revisions

1. Duration of the occlusion not mentioned.

Middle cerebral artery was occluded for 24 hours.

We mentioned the occlusion period in the following sentences.

Animals were assessed on neurobehavioral tests 24 h after surgery and brain tissues were collected for further study.” 

We added the sentence to make it easier to understand.

Middle cerebral artery was occluded for 24 h.

2. Interval between the surgery and testing if 24 hours is short.

Response:

Thank you for your kind review. Animal models of MCAO were used to induce cerebral ischemia, and the middle cerebral artery was blocked for 24 hours. This surgical method has been used in previous studies. We have previously reported the damage of cerebral cortex due to MCA occlusion for 24 hours. We think that this experimental model is suitable for the study of the neuroprotective effects of retinoic acid. We have already reported several papers demonstrating the neuroprotective effects of retinoic acid using this experimental model (Kang et al., 2021; 2022a; 2022b; 2023a;2023b)

Kang JB, Park DJ, Shah MA, Koh PO. Retinoic acid exerts neuroprotective effects against focal cerebral ischemia by preventing apoptotic cell death. Neurosci Lett. 2021. 757:135979. 

Kang JB, Koh PO. Identification of changed proteins by retinoic acid in cerebral ischemic damage: a proteomic study. J Vet Med Sci. 2022a. 84(9):1194-1204.

Kang JB, Shah MA, Park DJ, Koh PO. Retinoic acid regulates the ubiquitin-proteasome system in a middle cerebral artery occlusion animal model. Lab Anim Res. 2022b. 38(1):13.

Kang JB, Koh PO. Retinoic Acid Has Neuroprotective effects by modulating thioredoxin in ischemic brain damage and glutamate-exposed neurons. Neuroscience. 2023a. 521:166-181. 

Kang JB, Park DJ, Koh PO. Retinoic acid prevents the neuronal damage through the regulation of parvalbumin in an ischemic stroke model. Neurochem Res. 2023b. 48(2):487-501. 

We cited the references 

Middle cerebral artery was occluded for 24 h [35, 36].

We added the references

35. Kang JB, Shah MA, Park DJ, Koh PO. Retinoic acid regulates the ubiquitin-proteasome system in a middle cerebral artery occlusion animal model. Lab Anim Res. 2022;38: 13.

36. Kang JB, Park DJ, Shah MA, Koh PO. Retinoic acid exerts neuroprotective effects against focal cerebral ischemia by preventing apoptotic cell death. Neurosci Lett. 2021;757: 135979

3. How did the authors avoid bias in their scoring? 

Response:

Thank you for your kind and exact comments. 

Neurobehavioral deficits were assessed by the neurological deficits scoring test (Hattori et al., 2000). Neurobehavioral deficits were scored on a five-point scale according to the observation of behavioral changes as follows: no recognizable neurological deficits (0); lack of spontaneous motor activity or flexion of the contralate

---

## [Decision Letter · Decision Letter 1]

8 Jan 2024

PONE-D-23-19266R1Retinoic acid alleviates the reduction of Akt and Bad phosphorylation and regulates Bcl-2 family protein interactions in animal models of ischemic strokePLOS ONE

Dear Dr. Koh,

Thank you for submitting your manuscript to PLOS ONE. After careful consideration, we feel that it has merit but does not fully meet PLOS ONE’s publication criteria as it currently stands. Therefore, we invite you to submit a revised version of the manuscript that addresses the points raised during the review process.

We look forward to receiving your revised manuscript.

Kind regards,

Ahmed E. Abdel Moneim

Academic Editor

PLOS ONE

Journal Requirements:

Reviewers' comments:

Reviewer's Responses to Questions

**Comments to the Author**

1. If the authors have adequately addressed your comments raised in a previous round of review and you feel that this manuscript is now acceptable for publication, you may indicate that here to bypass the “Comments to the Author” section, enter your conflict of interest statement in the “Confidential to Editor” section, and submit your "Accept" recommendation.

Reviewer #1: All comments have been addressed

Reviewer #2: All comments have been addressed

2. Is the manuscript technically sound, and do the data support the conclusions?

Reviewer #1: Yes

Reviewer #2: Yes

3. Has the statistical analysis been performed appropriately and rigorously? 

Reviewer #1: Yes

Reviewer #2: Yes

4. Have the authors made all data underlying the findings in their manuscript fully available?

Reviewer #1: Yes

Reviewer #2: Yes

5. Is the manuscript presented in an intelligible fashion and written in standard English?

Reviewer #1: Yes

Reviewer #2: Yes

6. Review Comments to the Author

Reviewer #1: 1. Middle cerebral artery occlusion surgery section in methodology is repeated.

2. How many brain slices were mounted on glass slides for Cresyl Violet and TUNEL staining? How are these slices selected? was it random or after specific successions?

3. Line 39 there is double period

Reviewer #2: The author has provided a detailed response to the comments, and I recommend acceptance of the manuscript.

7. PLOS authors have the option to publish the peer review history of their article (what does this mean?). If published, this will include your full peer review and any attached files.

Reviewer #1: No

Reviewer #2: **Yes: **Yang Xu

---

## [Author Response · Author response to Decision Letter 1]

9 Jan 2024

Reviewer #1: Major Revisions

The authors showed that retinoic acid pretreatment can alleviate stroke induced by middle cerebral artery occlusion. They showed this by neurological scoring and other tests with histopathological changes and western blot analysis for protein expressions.

1. The author fails to show a direct mechanism in which retinoic acid is exerting the effect observed.

Response:

We appreciate for your kind comments. We tried to answer your questions very carefully.

Retinoic acid has several properties such as anti-oxidation (Ahlemeyer., 2001), anti-inflammation (Oliveira et al., 2018), anti-cancer (Hunsu et at., 2021). Especially in the brain, retinoic acid has neuroprotective effects in neurodegenerative diseases such as amyotrophic lateral sclerosis (Zhu et al., 2020), Parkinson's disease (Esteves et al., 2015) and ischemic stroke (Cai et al., 2019). In this study, we focused on the specific protective mechanism of retinoic acid on cerebral ischemia. The signal pathways involved in stroke is very diverse and complex. The regulatory pathways of retinoic acid in ischemic strokes are also diverse and complex. Among these multiple pathways, we focused on the PI3K/Akt pathway, which is a signal transduction pathway that promotes survival and growth in response to extracellular signals. Akt pathway regulates glucose metabolism, protein synthesis, mitochondrial metabolism, lipid metabolism, angiogenesis, autophagy, proliferation, and cell growth (Avan et al., 2016). It is a representative pathway related to cell survival. The regulation of Akt pathway by retinoic acid in stroke has not been reported much. Akt inhibits apoptosis through phosphorylation of proapoptotic proteins such as Bad and forkhead transcription factor (FKHR). However, dephosphorylated Bad forms heterodimers with Bcl-2 and Bcl-xL to inactivate them, allowing apoptosis. We showed that retinoic acid treatment alleviates MCAO-induced decreases in phospho-PDK1, phospho-Akt and phospho-Bad expression. Retinoic acid attenuated decrease in interaction between phospho-Bad and 14-3-3 caused by MCAO damage. Furthermore, retinoic acid mitigated the increase in Bcl-2/Bad and Bcl-xL/Bad binding levels and the reduction in Bcl-2/Bax and Bcl-xL/Bax binding levels caused by MCAO damage. Retinoic acid alleviated MCAO-induced increase of caspase-3 and cleaved caspase-3 expression. We demonstrate that retinoic acid prevented apoptosis against cerebral ischemia through phosphorylation of Akt and Bad, maintenance of phospho-Bad and 14-3-3 binding, and regulation of Bcl-2 family protein interactions. Retinoic acid exerts neuroprotective effects through various protective mechanisms during brain injury. However, it is difficult to discuss all protective mechanism of retinoic acid in this study. Among these various mechanisms, we focused on the PI3K/Akt signal pathway, which is a representative survival pathway. Previous studies demonstrated that retinoic acid activated the Akt signaling pathway and promoted cell survival in various tissues. However, data on the neuroprotective effects of retinoic acid on the PI3K-Akt signaling pathway in cerebral ischemia are limited. It has not been previously reported whether retinoic acid regulates the expression of phospho-Bad and the interaction between phospho-Bad and 14-3-3 in a stroke animal model. Therefore, we focused on the regulatory mechanism of Akt and its downstream target Bad by retinoic acid in cerebral ischemia. We regret that this study did not show various neuroprotective mechanism of retinoic acid in stroke animal model. However, we clearly showed that retinoic acid is involved in neuroprotection through the regulation of Akt and its downstream targets Bad in ischemic brain injury. We suggest that various protective mechanisms of retinoic acid should be researched. Thus, we plan to study other mechanisms in future research with reference to your kind comments.

References

Ahlemeyer B, Bauerbach E, Plath M, Steuber M, Heers C, Tegtmeier F, et al. Retinoic acid reduces apoptosis and oxidative stress by preservation of SOD protein level. Free Radic Biol Med. 2001;30(10):1067-77. 

Oliveira LM, Teixeira FME, Sato MN. Impact of retinoic acid on immune cells and inflammatory diseases. Mediators Inflamm. 2018;2018:3067126.

Hunsu VO, Facey COB, Fields JZ, Boman BM. Retinoids as chemo-preventive and molecular-targeted anti-cancer therapies. International Journal of Molecular Sciences. 2021; 22(14):7731. 

Zhu Y, Liu Y, Yang F, Chen W, Jiang J, He P, et al. All-trans retinoic acid exerts neuroprotective effects in amyotrophic lateral sclerosis-like Tg (SOD1*G93A)1Gur mice. Mol Neurobiol. 2020;57(8):3603-3615.

Esteves M, Cristóvão AC, Saraiva T, Rocha SM, Baltazar G, Ferreira L, et al. Retinoic acid-loaded polymeric nanoparticles induce neuroprotection in a mouse model for Parkinson's disease. Front Aging Neurosci. 2015;7:20.

Cai W, Wang J, Hu M, Chen X, Lu Z, Bellanti JA, et al. All trans-retinoic acid protects against acute ischemic stroke by modulating neutrophil functions through STAT1 signaling. J Neuroinflammation. 2019;16(1):175.

Avan A, Narayan R, Giovannetti E, Peters GJ. Role of Akt signaling in resistance to DNA-targeted therapy. World J Clin Oncol. 2016;7(5):352-369.

We added a sentence to the discussion section to make it easier to understand.

Retinoic acid exerts neuroprotective effects through various protective mechanisms during brain injury. However, it is difficult to discuss all protective mechanism of retinoic acid in this study. We focused on the regulatory mechanism of Akt and its downstream target Bad by retinoic acid in cerebral ischemia. We clearly showed that retinoic acid is involved in neuroprotection through the regulation of Akt and Bad in ischemic brain injury. We suggest that various protective mechanisms of retinoic acid should be researched. 

2. The signaling pathway observed is not the only major pathway involved in stroke, other pathways have been implicated as well. Moreso, retinoic acid not only activates Akt/PI3K pathway alone. The changes in other stroke pathways in relation to retinoic acid should also be investigated.

Response: 

We appreciate for your kind comments

Ischemic stroke is caused primarily by an interruption of cerebral blood flow, causes severe nerve damage, and is one of the leading causes of death and disability worldwide. It induces cell excitotoxicity, mitochondrial dysfunction, neuroinflammation, blood-brain barrier damage, and apoptotic processes. It is also known that the mechanism of causing ischemic stroke is very complicated. Moreover, the signaling pathway of ischemic stroke is very diverse and complex. The typical signal pathway involved in stroke is as follows; phosphatidylinositol 3-kinase (PI3K)-Akt signaling pathway, phosphatase and tensin homolog (PTEN) signaling pathway, death-associated protein kinase 1 (DAPK1) signaling pathway, postsynaptic density protein-95 (PSD95)/neuronal nitric oxide synthase (nNOS) signaling pathways, hypoxia-inducible factor (HIF) signaling pathway, nuclear factor E2-related factor 2 (Nrf2) signaling pathway, casein kinase 2 (CK2) signaling pathway, mTOR-related signaling pathways, and p53-mediated apoptotic pathway (Jo et al., 2012, Zhang et al., 2013, Shamloo et al., 2005, Guo et al., 2009, Dinkova-Kostova and Abramov, 2015, Bastian et al., 2019, Perez-Alvarez et al., 2018, Leker et al., 2004). As mentioned above, stroke is associated with various signaling pathways, but it is difficult to discuss all pathways in this study. Among these various signaling pathways, we focused on the PI3K/Akt signal pathway, which is a representative survival pathway. We investigated the change of Akt and its downstream target Bad by retinoic acid in cerebral ischemia.

Reference

Jo H, Mondal S, Tan D, Nagata E, Takizawa S, Sharma AK, et al. Small molecule-induced cytosolic activation of protein kinase Akt rescues ischemia-elicited neuronal death. Proc Natl Acad Sci USA. 2012;109: 10581-10586.

Zhang S, Taghibiglou C, Girling K, Dong Z, Lin SZ, Lee W, et al. Critical role of increased PTEN nuclear translocation in excitotoxic and ischemic neuronal injuries. J Neurosci. 2013;33: 7997-8008.

Shamloo M, Soriano L, Wieloch T, Nikolich K, Urfer R, Oksenberg D. Death-associated protein kinase is activated by dephosphorylation in response to cerebral ischemia. J Biol Chem. 2005;280: 42290-4229.

Guo S, Miyake M, Liu KJ, Shi H. Specific inhibition of hypoxia inducible factor 1 exaggerates cell injury induced by in vitro ischemia through deteriorating cellular redox environment. J Neurochem. 209;108: 1309-1321.

Dinkova-Kostova AT, Abramov AY. The emerging role of Nrf2 in mitochondrial function. Free Radic Biol Med. 2015;88: 179-188.

Bastian C, Quinn J, Tripathi A, Aquila D, McCray A, Dutta R, et al. CK2 inhibition confers functional protection to young and aging axons against ischemia by differentially regulating the CDK5 and AKT signaling pathways. Neurobiol Dis. 2019;126: 47-61.

Perez-Alvarez MJ, Villa Gonzalez M, Benito-Cuesta I, Wandosell FG. Role of mTORC1 controlling proteostasis after brain ischemia. Front Neurosci. 2018;12: 60.

Leker RR, Aharonowiz M, Greig NH, Ovadia H. The role of p53-induced apoptosis in cerebral ischemia: effects of the p53 inhibitor pifithrin alpha. Exp Neurol. 2004;187: 478-486.

In addition, retinoic acid plays a variety of roles including anti-oxidation, anti-inflammation, and anti-cancer., Retinoic acid inhibited cell death through regulation of apoptosis-related proteins (Kang et al., 2021). It regulates calcium concentration through the regulation of calcium binding proteins and inhibits apoptosis in response to brain ischemic injury (Kang et al., 2023). It also modulated MAP kinase pathway and caspase cascade. In previous studies, retinoic acid activated the Akt signaling pathway and promoted cell survival in various tissues. However, data on the neuroprotective effects of retinoic acid on the PI3K-Akt signaling pathway in cerebral ischemia are limited. It has not been previously reported whether retinoic acid regulates the expression of phospho-Bad and the interaction between phospho-Bad and 14-3-3 in a stroke animal model. Therefore, this study was designed to investigate the neuroprotective effects of retinoic acid on cerebral ischemia and the regulation of phospho-Akt and phospho-Bad by retinoic acid. In addition, we examined whether retinoic acid inhibits apoptosis and protects brain tissues from cerebral ischemia by controlling the binding of phospho-Bad with 14-3-3 and binding of Bcl-2 and Bcl-xL to Bad or Bax. Therefore, we focused on the regulatory mechanism of AKT and its downstream target Bad by retinoic acid in cerebral ischemia. 

We added sentences to introduction section to make it easier to understand.

Ischemic stroke induces cell excitotoxicity, mitochondrial dysfunction, blood-brain barrier damage, neuroinflammation, and apoptotic processes. It is also known that the mechanism of causing ischemic stroke is very complicated. Moreover, the signaling pathway of ischemic stroke is very diverse and complex. The signal pathways involved in stroke are PI3K/Akt signaling pathway, phosphatase and tensin homolog signaling pathway, death-associated protein kinase 1 signaling pathway, neuronal nitric oxide synthase signaling pathways, hypoxia-inducible factor signaling pathway, nuclear factor E2-related factor 2 signaling pathway, casein kinase 2 signaling pathway, mTOR-related signaling pathways, and p53-mediated apoptotic pathway [23-30]. As mentioned above, stroke is associated with various signaling pathways, but it is difficult to discuss all pathways in this study. Among these various signaling pathways, we focused on the PI3K/Akt signal pathway, which is a representative survival pathway

We added references to reference list.

23. Jo H, Mondal S, Tan D, Nagata E, Takizawa S, Sharma AK, et al. Small molecule-induced cytosolic activation of protein kinase Akt rescues ischemia-elicited neuronal death. Proc Natl Acad Sci USA. 2012;109: 10581-10586.

24. Zhang S, Taghibiglou C, Girling K, Dong Z, Lin SZ, Lee W, et al. Critical role of increased PTEN nuclear translocation in excitotoxic and ischemic neuronal injuries. J Neurosci. 2013;33: 7997-8008.

25. Shamloo M, Soriano L, Wieloch T, Nikolich K, Urfer R, Oksenberg D. Death-associated protein kinase is activated by dephosphorylation in response to cerebral ischemia. J Biol Chem. 2005;280: 42290-4229.

26. Guo S, Miyake M, Liu KJ, Shi H. Specific inhibition of hypoxia inducible factor 1 exaggerates cell injury induced by in vitro ischemia through deteriorating cellular redox environment. J Neurochem. 209;108: 1309-1321.

27. Dinkova-Kostova AT, Abramov AY. The emerging role of Nrf2 in mitochondrial function. Free Radic Biol Med. 2015;88: 179-188.

28. Bastian C, Quinn J, Tripathi A, Aquila D, McCray A, Dutta R, et al. CK2 inhibition confers functional protection to young and aging axons against ischemia by differentially regulating the CDK5 and AKT signaling pathways. Neurobiol Dis. 2019;126: 47-61.

29. Perez-Alvarez MJ, Villa Gonzalez M, Benito-Cuesta I, Wandosell FG. Role of mTORC1 controlling proteostasis after brain ischemia. Front Neurosci. 2018;12: 60.

30. Leker RR, Aharonowiz M, Greig NH, Ovadia H. The role of p53-induced apoptosis in cerebral ischemia: effects of the p53 inhibitor pifithrin alpha. Exp Neurol. 2004;187: 478-486.

Minor revisions

1. Duration of the occlusion not mentioned.

Middle cerebral artery was occluded for 24 hours.

We mentioned the occlusion period in the following sentences.

Animals were assessed on neurobehavioral tests 24 h after surgery and brain tissues were collected for further study.” 

We added the sentence to make it easier to understand.

Middle cerebral artery was occluded for 24 h.

2. Interval between the surgery and testing if 24 hours is short.

Response:

Thank you for your kind review. Animal models of MCAO were used to induce cerebral ischemia, and the middle cerebral artery was blocked for 24 hours. This surgical method has been used in previous studies. We have previously reported the damage of cerebral cortex due to MCA occlusion for 24 hours. We think that this experimental model is suitable for the study of the neuroprotective effects of retinoic acid. We have already reported several papers demonstrating the neuroprotective effects of retinoic acid using this experimental model (Kang et al., 2021; 2022a; 2022b; 2023a;2023b)

Kang JB, Park DJ, Shah MA, Koh PO. Retinoic acid exerts neuroprotective effects against focal cerebral ischemia by preventing apoptotic cell death. Neurosci Lett. 2021. 757:135979. 

Kang JB, Koh PO. Identification of changed proteins by retinoic acid in cerebral ischemic damage: a proteomic study. J Vet Med Sci. 2022a. 84(9):1194-1204.

Kang JB, Shah MA, Park DJ, Koh PO. Retinoic acid regulates the ubiquitin-proteasome system in a middle cerebral artery occlusion animal model. Lab Anim Res. 2022b. 38(1):13.

Kang JB, Koh PO. Retinoic Acid Has Neuroprotective effects by modulating thioredoxin in ischemic brain damage and glutamate-exposed neurons. Neuroscience. 2023a. 521:166-181. 

Kang JB, Park DJ, Koh PO. Retinoic acid prevents the neuronal damage through the regulation of parvalbumin in an ischemic stroke model. Neurochem Res. 2023b. 48(2):487-501. 

We cited the references 

Middle cerebral artery was occluded for 24 h [35, 36].

We added the references

35. Kang JB, Shah MA, Park DJ, Koh PO. Retinoic acid regulates the ubiquitin-proteasome system in a middle cerebral artery occlusion animal model. Lab Anim Res. 2022;38: 13.

36. Kang JB, Park DJ, Shah MA, Koh PO. Retinoic acid exerts neuroprotective effects against focal cerebral ischemia by preventing apoptotic cell death. Neurosci Lett. 2021;757: 135979

3. How did the authors avoid bias in their scoring? 

Response:

Thank you for your kind and exact comments. 

Neurobehavioral deficits were assessed by the neurological deficits scoring test (Hattori et al., 2000). Neurobehavioral deficits were scored on a five-point scale according to the observation of behavioral changes as follows: no recognizable neurological deficits (0); lack of spontaneous motor activity or flexion of the contralate

---

## [Decision Letter · Decision Letter 2]

5 Mar 2024

PONE-D-23-19266R2Retinoic acid alleviates the reduction of Akt and Bad phosphorylation and regulates Bcl-2 family protein interactions in animal models of ischemic strokePLOS ONE

Dear Dr. Koh,

Thank you for submitting your manuscript to PLOS ONE. After careful consideration, we feel that it has merit but does not fully meet PLOS ONE’s publication criteria as it currently stands. Therefore, we invite you to submit a revised version of the manuscript that addresses the points raised during the review process.

We look forward to receiving your revised manuscript.

Kind regards,

Ahmed E. Abdel Moneim

Academic Editor

PLOS ONE

Reviewers' comments:

Reviewer's Responses to Questions

**Comments to the Author**

1. If the authors have adequately addressed your comments raised in a previous round of review and you feel that this manuscript is now acceptable for publication, you may indicate that here to bypass the “Comments to the Author” section, enter your conflict of interest statement in the “Confidential to Editor” section, and submit your "Accept" recommendation.

Reviewer #1: (No Response)

Reviewer #4: (No Response)

Reviewer #5: (No Response)

Reviewer #6: (No Response)

2. Is the manuscript technically sound, and do the data support the conclusions?

Reviewer #1: Yes

Reviewer #4: Partly

Reviewer #5: Partly

Reviewer #6: Yes

3. Has the statistical analysis been performed appropriately and rigorously? 

Reviewer #1: Yes

Reviewer #4: No

Reviewer #5: Yes

Reviewer #6: Yes

4. Have the authors made all data underlying the findings in their manuscript fully available?

Reviewer #1: Yes

Reviewer #4: No

Reviewer #5: Yes

Reviewer #6: Yes

5. Is the manuscript presented in an intelligible fashion and written in standard English?

Reviewer #1: Yes

Reviewer #4: No

Reviewer #5: No

Reviewer #6: Yes

6. Review Comments to the Author

Reviewer #1: Last comments were not addressed. I don't know if the editor did not get across to you my minor comments. The MCAO surgery protocol was written twice in the methodology section, there was 2 full stops after a sentence in line 39 and the method of election of brain slices used for CFV and TUNEL staining is still missing.

Reviewer #4: Dear Author,

Thanks for your manuscript submission.

1. Clarity and Organization:

- The overall structure of the manuscript needs improvement for clarity.

- The introduction should provide more context on retinoic acid, ischemic stroke, and the PI3K-Akt pathway.

- Consider providing a clearer rationale for choosing the specific doses and duration of retinoic acid administration.

2. Background and Literature Review:

- The introduction lacks a comprehensive review of relevant literature. It should cover existing knowledge on retinoic acid, its role in stroke, and the connection with the PI3K-Akt pathway.

- The significance of the PI3K-Akt pathway in stroke needs more emphasis, along with recent developments in the field.

3. Methodology:

- The methods section needs more details, especially regarding the experimental design, animal model characteristics, and surgical procedures.

- Specify the criteria for selecting the 5 mg/kg dose of retinoic acid and the rationale for the four-day treatment period.

- Provide information on the choice of neurobehavioral tests used and justify why these specific tests were selected.

4. Results:

- The results section lacks clarity in presenting the findings. Consider reorganizing and presenting the results in a more logical sequence.

- Include statistical analyses and significance levels for the observed changes.

- Clarify the timeline of events and measurements, especially in relation to the administration of retinoic acid and the induction of ischemic stroke.

5. Discussion:

- The discussion needs to be more thorough and should interpret the results in the context of existing literature.

- Provide a detailed explanation of how retinoic acid affects the PI3K-Akt pathway and its downstream proteins.

- Discuss potential limitations of the study and avenues for future research.

- Address the clinical relevance of the findings and potential implications for stroke therapy.

6. Figures and Tables:

- Ensure that figures and tables are clear and easy to interpret.

- Include proper labeling and legends for all figures and tables.

- Consider whether additional figures or tables could enhance the presentation of results.

7. Language and Style:

- Improve the overall writing style for clarity and precision.

- Check grammar and sentence structure throughout the manuscript.

- Eliminate unnecessary repetition and jargon that may hinder understanding for readers outside the specific field.

8. Conclusion:

- The conclusion should be strengthened, summarizing the key findings and their implications more clearly.

- Avoid introducing new information in the conclusion; focus on summarizing and emphasizing the main contributions of the study.

Reviewer #5: Before the final comments on this manuscript, the author needs to address the comments in a scientific manner.

Reviewer #6: The paper written by Ju-Bin Kang et al. found that retinoic acid can effectively relieve the brain nerve injury caused by ischemic stroke. Overall, the study looks very interesting, but there are still some questions that need to be addressed:

1.Why were male rats chosen? Is there gender difference for ischemic stroke?

2.Why was retinoic acid injected four days before surgery?

3.Besides PI3K-Akt signaling pathway, what other pathways are involved in ischemic stroke?

4.How to understand that the grip strength of the left and right forelimbs was examined, but only one forelimb is used in each test?

5.Are there any references for supporting the content in Lines 57-59?

6.There are some grammatical errors in this manuscript, and it is better to find a native speaker to correct it.

7.In the Reference, please check them carefully and modify them according to the requirements of the Journal.

7. PLOS authors have the option to publish the peer review history of their article (what does this mean?). If published, this will include your full peer review and any attached files.

Reviewer #1: No

Reviewer #4: **Yes: **Sidharth Mehan

Reviewer #5: **Yes: **Sumit Kumar

Reviewer #6: No

---

## [Author Response · Author response to Decision Letter 2]

5 Apr 2024

Reviewer #1: 

Last comments were not addressed. I don't know if the editor did not get across to you my minor comments. The MCAO surgery protocol was written twice in the methodology section, there was 2 full stops after a sentence in line 39 and the method of election of brain slices used for CFV and TUNEL staining is still missing.

Response:

Thank you for your valuable comment and thought review for our manuscript. We removed the duplicated section. We used the brain range from bregma levels +2.0 mm to -2.0 mm for further research [Chen SH et al., 2002].

References

Chen SH, Fung PC, Cheung RT. Neuropeptide Y-Y1 receptor modulates nitric oxide level during stroke in the rat. Free Radic Biol Med. 2002;32: 776-784. 

We inserted the sentence in materials and methods section.

 Brain tissues from bregma levels +2.0 mm to -2.0 mm were fixed with 4% paraformaldehyde solution and washed with tap water overnight to remove paraformaldehyde. They were dehydrated with the ethyl alcohol series from 70% to 100% and washed with xylene. 

Reviewer #4: 

Thanks for your manuscript submission.

1. Clarity and Organization:

- The overall structure of the manuscript needs improvement for clarity. The introduction should provide more context on retinoic acid, ischemic stroke, and the PI3K-Akt pathway. 

Response:

Thank you for your valuable comment and thought review for our manuscript. Retinoic acid has been widely studied for its regulatory mechanism in various cellular processes, including tissue damage and recovery. Retinoic acid treatment modulates the expression of key molecules involved in apoptosis such as PI3K, Akt, Bad, and caspase-3 [Jiang et al., 2018]. Retinoic acid has also been shown to enhance neural differentiation by upregulating DAX1 levels through the PI3K-Akt pathway [Nagl et al., 2009]. Ischemic stroke is a complex neurological disorder in which signaling pathways are disrupted [Wang et al., 2020]. PI3K-Akt pathway is a representative signaling pathway involved in ischemic stroke [Wang et al., 2020]. Moreover, recent study has focused on the potential therapeutic significance through PI3K-Akt pathway activation in ischemic stroke [Gu et al., 2022].

References

Jiang W, Guo M, Gong M, Chen L, Bi Y, Zhang Y, Shi Y, Qu P, Liu Y, Chen J, Li T. Vitamin A bio-modulates apoptosis via the mitochondrial pathway after hypoxic-ischemic brain damage. Mol Brain. 2018;13: 11:14. 

Nagl F, Schönhofer K, Seidler B, Mages J, Allescher HD, Schmid RM, Schneider G, Saur D. Retinoic acid-induced nNOS expression depends on a novel PI3K/Akt/DAX1 pathway in human TGW-nu-I neuroblastoma cells. Am J Physiol Cell Physiol. 2009;297: C1146-56.

Wang MM, Zhang M, Feng YS, Xing Y, Tan ZX, Li WB, Dong F, Zhang F. Electroacupuncture inhibits neuronal autophagy and apoptosis via the PI3K/AKT pathway following ischemic stroke. Front Cell Neurosci. 2020;15: 14:134. 

Gu C, Zhang Q, Li Y, Li R, Feng J, Chen W, Ahmed W, Soufiany I, Huang S, Long J, Chen L. The PI3K/AKT pathway-the potential key mechanisms of traditional chinese medicine for stroke. Front Med (Lausanne). 2022;31: 9:900809. 

We revised and inserted the sentences the introduction part .

In previous studies, retinoic acid activated the Akt signaling pathway and promoted cell survival in various tissues [5, 33, 34]. It has also been shown to enhance neural differentiation by upregulating DAX1 levels through the PI3K-Akt pathway [35]. Ischemic stroke is a complex neurological disorder in which signaling pathways are disrupted [36]. PI3K-Akt pathway is a representative signaling pathway involved in ischemic stroke [36]. Moreover, recent study has focused on the potential therapeutic significance through PI3K-Akt pathway activation in ischemic stroke [37].

We inserted the references in reference section. 

35. Nagl F, Schönhofer K, Seidler B, Mages J, Allescher HD, Schmid RM, Schneider G, Saur D. Retinoic acid-induced nNOS expression depends on a novel PI3K/Akt/DAX1 pathway in human TGW-nu-I neuroblastoma cells. Am J Physiol Cell Physiol. 2009;297: C1146-56.

36. Wang MM, Zhang M, Feng YS, Xing Y, Tan ZX, Li WB, Dong F, Zhang F. Electroacupuncture inhibits neuronal autophagy and apoptosis via the PI3K/AKT pathway following ischemic stroke. Front Cell Neurosci. 2020;15: 14:134.

37. Gu C, Zhang Q, Li Y, Li R, Feng J, Chen W, Ahmed W, Soufiany I, Huang S, Long J, Chen L. The PI3K/AKT Pathway-The Potential Key Mechanisms of Traditional Chinese Medicine for Stroke. Front Med (Lausanne). 2022;9: 900809.

- Consider providing a clearer rationale for choosing the specific doses and duration of retinoic acid administration.

Response:

We really appreciate your careful review and kind comments. We decided the dose of retinoic acid (5 mg/kg) by referring to the previously reported method [Kong et al., 2015]. The experimental animals were treated with 0.2, 1, 5, 25 mg/kg of retinoic acid to determine the protective effect of retinoic acid. Retinoic acid attenuates blood-brain barrier (BBB) permeability and infarction caused by MCAO operation. They showed that BBB-specific genes are significantly increased after treatment with either 5 or 25 mg/kg of retinoic acid compared with the vehicle group. They reported a neuroprotective effect on at least 5 mg/kg of retinoic acid. Based on the previous study, the dose of retinoic acid was determined to be 5 mg/kg, and the neuroprotective effect of retinoic acid was confirmed at this dose [Kang et al., 2021]. 

Kong L, Wang Y, Wang XJ, Wang XT, Zhao Y, Wang LM, et al. Retinoic acid ameliorates blood-brain barrier disruption following ischemic stroke in rats. Pharmacol Res. 2015;99: 125-136.

Kang JB, Park DJ, Shah MA, Koh PO. Retinoic acid exerts neuroprotective effects against focal cerebral ischemia by preventing apoptotic cell death. Neurosci Lett. 2021;757:135979. 

We inserted the sentence in materials and methods

Animals were treated with 5 mg/kg of retinoic acid, and the dose and duration of treatment of retinoic acid were determined by the previously described method [38]. We previously confirmed the neuroprotective effect of retinoic acid at this dose and duration of retinoic acid administration [39]. 

We revised the references in reference list

38. Kong L, Wang Y, Wang XJ, Wang XT, Zhao Y, Wang LM, et al. Retinoic acid ameliorates blood-brain barrier disruption following ischemic stroke in rats. Pharmacol Res. 2015;99: 125-136.

293.

39. Kang JB, Shah MA, Park DJ, Koh PO. Retinoic acid regulates the ubiquitin-proteasome system in a middle cerebral artery occlusion animal model. Lab Anim Res. 2022;38: 13.

2. Background and Literature Review:

- The introduction lacks a comprehensive review of relevant literature. It should cover existing knowledge on retinoic acid, its role in stroke, and the connection with the PI3K-Akt pathway. The significance of the PI3K-Akt pathway in stroke needs more emphasis, along with recent developments in the field.

Response:

We revised and inserted the sentences the introduction part .

In previous studies, retinoic acid activated the Akt signaling pathway and promoted cell survival in various tissues [5, 33, 34]. It has also been shown to enhance neural differentiation by upregulating DAX1 levels through the PI3K-Akt pathway [35]. Ischemic stroke is a complex neurological disorder in which signaling pathways are disrupted [36]. PI3K-Akt pathway is a representative signaling pathway involved in ischemic stroke [36]. Moreover, recent study has focused on the potential therapeutic significance through PI3K-Akt pathway activation in ischemic stroke [37].

We inserted the references in reference section. 

35. Nagl F, Schönhofer K, Seidler B, Mages J, Allescher HD, Schmid RM, Schneider G, Saur D. Retinoic acid-induced nNOS expression depends on a novel PI3K/Akt/DAX1 pathway in human TGW-nu-I neuroblastoma cells. Am J Physiol Cell Physiol. 2009;297: C1146-56.

36. Wang MM, Zhang M, Feng YS, Xing Y, Tan ZX, Li WB, Dong F, Zhang F. Electroacupuncture inhibits neuronal autophagy and apoptosis via the PI3K/AKT pathway following ischemic stroke. Front Cell Neurosci. 2020;15: 14:134.

37. Gu C, Zhang Q, Li Y, Li R, Feng J, Chen W, Ahmed W, Soufiany I, Huang S, Long J, Chen L. The PI3K/AKT Pathway-The Potential Key Mechanisms of Traditional Chinese Medicine for Stroke. Front Med (Lausanne). 2022;9: 900809.

3. Methodology:

- The methods section needs more details, especially regarding the experimental design, animal model characteristics, and surgical procedures.

Response:

We appreciate your careful review and kind comments. As you recommended, we inserted more details about experimental animal design and procedure.

We revised and inserted the sentences in materials and methods section.

Experimental animal preparation

Male Sprague Dawley (n=40, 210-220 g) rats were obtained from Samtako Co. (Animal Breeding Center, Osan, Korea). Male animals were used to eliminate potential confounding variables associated with sex hormones. All experimental procedures were conducted by following approved guidelines of the Institutional Animal Care and Use Committee of Gyeongsang National University (Approval number: GNU-190218-R0008). Animals were housed with controlled temperature condition with 25℃ and lighting condition with 12 h light and 12 h dark cycle. They were randomly divided into four groups as follows: vehicle + sham, retinoic acid + sham, vehicle + middle cerebral artery occlusion (MCAO), and retinoic acid + MCAO. Retinoic acid (5 mg/kg, Sigma Aldrich, St. Louis, MO, USA) was dissolved in solvent agent (polyethylene glycol, 0.9% NaCl, and ethanol; 70%/20%/10% by volume) and injected via intraperitoneal cavity four days before surgery [38]. Animals were treated with 5 mg/kg of retinoic acid, and the dose and duration of treatment of retinoic acid were determined by the previously described method [38]. We previously confirmed the neuroprotective effect of retinoic acid at this dose and duration of retinoic acid administration [39]. Vehicle group animals were injected only solvent solutions. 

Middle cerebral artery occlusion surgery

MCAO surgery was performed to induce cerebral ischemia in the following method [40]. Animals were anesthetized by intraperitoneal injection with Zoletil (50 mg/kg, Virbac, Carros, France) 30 min after retinoic acid injection and kept in a heating pad in a supine position to maintain body temperature. Right common carotid artery (CCA) was exposed through the midline cervical incision and separated from adjacent tissues. External carotid artery (ECA) and internal carotid artery (ICA) were exposed. CCA was temporarily ligated using microvascular clamps, the laryngeal artery and cranial thyroid artery were resected, and the ECA was amputated. A 4/0 monofilament with heat-rounded tip was inserted into the cut ECA, continuously moved to the ICA until resistance was felt to block the origin of the middle cerebral artery, and ligated with ECA. Microvascular clips were removed and incised skin was sutured. Middle cerebral artery was occluded for 24 h [38, 39, 41]. Sham animals were operated the same procedure except for insertion of nylon filament. Neurobehavioral tests including neurological deficits scoring test, corner test, and grip strength test are commonly used in rodent models of stroke [42-44]. Animals were assessed on neurobehavioral tests 24 h after surgery. After anesthesia with Zoletil (Virbac), animals were quickly decapitated and sacrificed for experimentation. We tried to minimize pain to the animals and brain tissues were collected for further study.

We revised the refences.

42. Hattori K, Lee H, Hurn PD, Crain BJ, Traystman RJ, DeVries AC. Cognitive deficits after focal cerebral ischemia in mice. Stroke. 2000;31: 1939-1944. 

43. Ruan J, Yao Y. Behavioral tests in rodent models of stroke. Brain Hemorrhages. 2020;1: 171-184.

44. Takeshita H, Yamamoto K, Nozato S, Inagaki T, Tsuchimochi H, Shirai M, et al. Modified forelimb grip strength test detects aging-associated physiological decline in skeletal muscle function in male mice. Sci Rep. 2017;7: 42323. 

- Specify the criteria for selecting the 5 mg/kg dose of retinoic acid and the rationale for the four-day treatment period.

Response:

We appreciate your careful review and kind comments. We decided the dose of retinoic acid (5 mg/kg) by referring to the previously reported method [Kong et al., 2015]. The experimental animals were treated with 0.2, 1, 5, 25 mg/kg of retinoic acid to determine the protective effect of retinoic acid. Retinoic acid attenuates blood-brain barrier (BBB) permeability and infarction caused by MCAO operation. They showed that BBB-specific genes are significantly increased after treatment with either 5 or 25 mg/kg of retinoic acid compared with the vehicle group. They reported a neuroprotective effect on at least 5 mg/kg of retinoic acid. They also continuously treated retinoic acid 4 days before MCAO surgery, so we tried their method by referring to Kong et al. Actually, 5 mg/kg of retinoic acid was continuously administered to the experimental animals for 4 days before MCAO surgery, and we clearly identified the neuroprotective effect of retinoic acid at this dose and duration of retinoic acids [Kang et al., 2021]. 

References

Kong L, Wang Y, Wang XJ, Wang XT, Zhao Y, Wang LM, Chen ZY Retinoic acid ameliorates blood-brain barrier disruption following ischemic stroke in rats. Pharmacol Res. 2015;99: 125-136.

Kang JB, Park DJ, Shah MA, Koh PO. Retinoic acid exerts neuroprotective effects against focal cerebral ischemia by preventing apoptotic cell death. Neurosci Lett. 2021;757:135979. 

We modified and inserted the sentence in materials and methods

Animals were treated with 5 mg/kg of retinoic acid, and the dose and duration of treatment of retinoic acid were determined by the previously described method [38]. We previously confirmed the neuroprotective effect of retinoic acid at this dose and duration of retinoic acid administration [39].

- Provide information on the choice of neurobehavioral tests used and justify why these specific tests were selected.

Response:

Thank you for your kind review and comment. The neurological deficit score test, corner test, and grip strength test are motor/sensory motor and cognitive tests commonly used in rodent models of stroke [Ruan et al., 2020]. Neurological deficits scoring test is a combined assessment method that observes various neurological functions such as balance, coordination, reflexes, and sensory perception. The corner test observes a turning pattern towards the strong side during the corner turn. The grip strength test assesses motor function by measuring force. Behavioral tests were performed to confirm the behavioral deficits induced by MCAO and to investigate the effects of retinoic acid administration on behavioral improvement and neuroprotection. Neurological deficit score tests showed that retinoic acid-treated animals with MCAO damage scored lower than vehicle-treated animals with MCAO damage. In the corner test, retinoic acid-treated animals with MCAO damage showed a lower frequency of rotation in the same direction as the lesion, compared with vehicle-treated animal with MCAO damage. In the grip test, retinoic acid-treated animals with MCAO damage had a higher grip strength on the contralateral side compared to vehicle-treated animal with MCAO damage. These results confirmed that MCAO induces behavioral deficits and retinoic acid administration improves behavioral deficits. Therefore, we can demonstrate that retinoic acid has neuroprotective effects in focal cerebral ischemia.

References

Ruan J, Yao Y. Behavioral tests in rodent models of stroke. Brain Hemorrhages. 2020;1: 171-184. 

We inserted the sentence in materials and methods. 

Neurobehavioral tests including neurological deficits scoring test, corner test, and grip strength test are commonly used in rodent models of stroke [42-44]. Animals were assessed on neurobehavioral tests 24 h after surgery. 

We inserted the reference in reference section 

42. Ruan J, Yao Y. Behavioral tests in rodent models of stroke. Brain Hemorrhages. 2020;1: 171-184.

43. Hattori K, Lee H, Hurn PD, Crain BJ, Traystman RJ, DeVries AC. Cognitive deficits after focal cerebral ischemia in mice. Stroke. 2000

---

## [Decision Letter · Decision Letter 3]

22 Apr 2024

Retinoic acid alleviates the reduction of Akt and Bad phosphorylation and regulates Bcl-2 family protein interactions in animal models of ischemic stroke

PONE-D-23-19266R3

Dear Dr. Koh,

We’re pleased to inform you that your manuscript has been judged scientifically suitable for publication and will be formally accepted for publication once it meets all outstanding technical requirements.

Kind regards,

Ahmed E. Abdel Moneim

Academic Editor

PLOS ONE

Additional Editor Comments (optional):

Reviewers' comments:

Reviewer's Responses to Questions

**Comments to the Author**

1. If the authors have adequately addressed your comments raised in a previous round of review and you feel that this manuscript is now acceptable for publication, you may indicate that here to bypass the “Comments to the Author” section, enter your conflict of interest statement in the “Confidential to Editor” section, and submit your "Accept" recommendation.

Reviewer #1: All comments have been addressed

Reviewer #4: All comments have been addressed

Reviewer #6: (No Response)

2. Is the manuscript technically sound, and do the data support the conclusions?

Reviewer #1: Yes

Reviewer #4: Partly

Reviewer #6: Yes

3. Has the statistical analysis been performed appropriately and rigorously? 

Reviewer #1: Yes

Reviewer #4: Yes

Reviewer #6: Yes

4. Have the authors made all data underlying the findings in their manuscript fully available?

Reviewer #1: Yes

Reviewer #4: Yes

Reviewer #6: Yes

5. Is the manuscript presented in an intelligible fashion and written in standard English?

Reviewer #1: Yes

Reviewer #4: No

Reviewer #6: Yes

6. Review Comments to the Author

Reviewer #1: (No Response)

Reviewer #4: Dear author,

After careful revision, manuscript revised successfully, and can be proceed further for publication.

Reviewer #6: (No Response)

7. PLOS authors have the option to publish the peer review history of their article (what does this mean?). If published, this will include your full peer review and any attached files.

Reviewer #1: **Yes: **Azeez Ishola

Reviewer #4: No

Reviewer #6: No

---

## [Editor Report · Acceptance letter]

4 May 2024

PONE-D-23-19266R3 

PLOS ONE

Dear Dr. Koh, 

I'm pleased to inform you that your manuscript has been deemed suitable for publication in PLOS ONE. Congratulations! Your manuscript is now being handed over to our production team.

Kind regards, 

on behalf of

Dr. Ahmed E. Abdel Moneim 

Academic Editor

PLOS ONE